# Unbiased Decisions Reduce Regret: Adversarial Optimism for the Bank Loan Problem

## Abstract

In many real world settings binary classification decisions are made based on limited data in near real-time, e.g. when assessing a loan application. We focus on a class of these problems that share a common feature: the **true label is only observed when a data point is assigned a positive label** by the principal, e.g. we only find out if an applicant defaults if we *accept* their loan application in the first place. In this setting, sometimes referred to as the Bank Loan Problem (BLP) in the literature, the labelled training can accumulate bias since it is influenced by the past decisions. Prior work mitigates the consequences of this bias by injecting *optimism* into the model to allow the learner to correct self-reinforcing *false rejections*. This reduces long term regret but comes at the cost of a higher false acceptance rate. We introduce *adversarial optimism* (AdOpt) to directly address bias in the training set using *adversarial domain adaptation*. The goal of AdOpt is to learn an unbiased but informative representation of past data, by reducing the distributional shift between the set of *accepted* data points and all data points seen thus far. We integrate classification made using this "debiased" representation of the data with the recently proposed *pseudo-label optimism* (PLOT) method to increase the rate of correct decisions at every timestep. AdOpt significantly exceeds state-of-the-art performance on a set of challenging BLP benchmark problems.

## 1 Introduction

In a variety of online decision making problems, principals have to make an acceptance or rejection decision for a given instance based on observing data points in an online fashion. Across a broad range of these, the *true label* is only revealed for those data points which the principal accepted, creating a biased labelled dataset.

In this work we concentrate on addressing this issue for the specific class of binary classification tasks, also known as the "Bank Loan Problem"(BLP), motivated by the characteristic example of a lender deciding on outcomes of loan applications. The lender's objective is to maximize profit, i.e. accept as many credible applicants as possible, while denying those who would ultimately default. The caveat is that the lender doesn't learn whether rejected applicants would have actually repaid the loan. Hence a decision policy that relies solely on the past experience lacks the opportunity to correct for erroneous rejection decisions. These "false rejects" are self-reinforcing since the correct label is never revealed for rejected candidates.

The dynamic nature of the data collection mechanism in the BLP offers a very simple and clearly defined example of accumulating bias of the kind we indicated above. As time progresses, the growing pool of accepted applicants (the models training set), created by the models decisions, forms an increasingly biased dataset, whose distribution is different from that of the general applicant population. This distributional shift affects the accuracy of the predictions of a model trained on the set of accepted points for any future applicants.

A common approach for mitigating the consequences of a biased model is to inject *optimism* into the decision making strategy. Of particular note to us is the recently proposed Pseudo-Label Optimism (Pacchiano et al., 2021, PLOT) that provides a simple and computationally efficient way to introduce optimism which can be used in combination with deep neural networks (DNNs). This op-

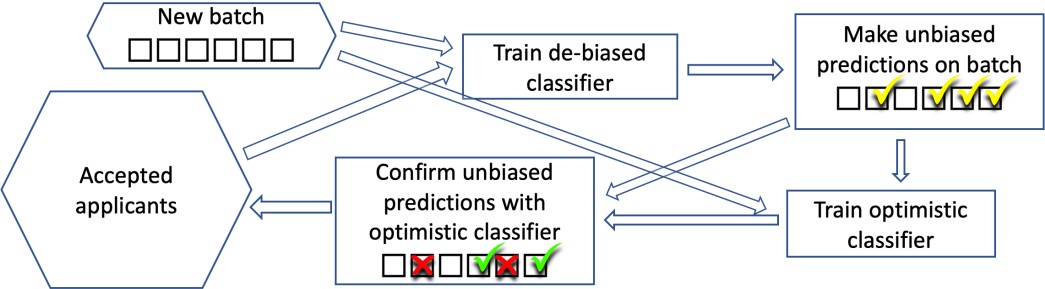

Figure 1: The AdOpt algorithm

timism translates *self-fulfilling* false rejects into *self-correcting* false accepts, which strictly reduces long term regret. However, the cost of optimism is an increased *false acceptance* rate in particular early on in the learning process.

We propose and evaluate a novel approach to the BLP that is motivated by methods from the domain of learning in the presence of *distributional shift*. Specifically, we utilize *adversarial domain adaptation* to learn a de-biased representation of the training data that minimizes this distributional difference, while preserving the informative features. Although very natural in this setting, this is to our knowledge the first attempt to utilise adversarial domain adaptation to tackle bias in the online context. Our experiments show that the de-biased classifier can achieve increased recall on the new queries while maintaining sufficient precision.

However, by itself this adversarially de-biased classifier suffers from a fundamental flaw: It needs to trade-off between *truly informative features* and reducing bias. Clearly, both cannot be accomplished at the same time, which leads to this method performing poorly in some settings.

To overcome these issues we introduce adversarial optimism method (AdOpt), which combines the de-biased classification approach with PLOT. When presented with a new query at each step, AdOpt uses a de-biased representation of the existing training data and trains a de-biased classifier to assess the probability of it being a true positive. If the de-biased classifier recommends to accept the point, we verify its suggestion using the *pseudo-label optimism* of PLOT: we add the data point to the original labelled dataset with a positive label and train the optimistic classifier on this mixed dataset. Finally, we use the classification of the optimistic classifier to accept or reject the data point (see Figure 1)

Compared with the PLOT, AdOpt has the advantage of utilizing the de-biased classifier for identification of candidates for the pseudo-label optimism routine. In Pacchiano et al. (2021) the pseudo-label optimism was combined with the $\epsilon$-*greedy* approach for pseudo-label candidates selection to mitigate the issue of high false-positive rate. However the major weakness of this approach is that the proportion of selected candidates for exploration stays constant from batch to batch, and that their selection is random and not in any way driven by the data. As a result re-running and carefully analyzing the experiments in the PLOT paper with several dataset sampling strategies shows that it doesn't consistently beat SOTA on 2 out of 3 datasets from Pacchiano et al. (2021)(see Figure 2 and section 5). The upshot is that better way of choosing pseudo-label candidates is needed to achieve optimal performance. Since the de-biased classifier is able to catch more positives with better accuracy, by combining it with PLOT we converge to better accuracy for the dataset after seeing a smaller number of candidates and with fewer misclassified queries.

We evaluate AdOpt against a number of established approaches from literature, namely PLOT, "greedy", $\epsilon$-greedy (with a decaying schedule) and NeuralUCB Zhou et al. (2020) algorithms. AdOpt outperforms the state-of-the-art methods on 4 out of 5 benchmark datasets. Conducting high number of experiments across several sampling methods allows us to use t-test to confirm the statistical significance of our results with high accuracy (Figures 2, 3, 4).

Our ablation study on the effectiveness of PLOT in the AdOpt algorithm demonstrates that the addition of PLOT significantly reduces the wide standard deviation that constitutes the main weakness of a standalone adversarially de-biased classifier (Figure 2). In addition, the Standalone Adversarial

approach requires hyperparameter tuning to achieve balance between recall and precision, whereas AdOpt performance is significantly less affected by this balance in our experiments.

The use of the de-biased classification routine has potential to reduce the risk of accumulating bias in online decision making which could have profound ramifications for a variety of socially relevant issues, such as access to funding or bail decisions. We have conducted a preliminary evaluation and summarised the results in Figure 5 of the Appendix that support this conjecture. This potential property of the adversarial approach is especially attractive since the increase in fairness occurs as a direct consequence of attempting to improve classification performance and not due to introducing any kind of artificial constraint, e.g. in the form of a "protected" characteristic. We delay the thorough investigation of this direction to future work.

## 2 RELATED WORK

**Contextual bandits and function approximation**  The Bank Loan problem (BLP) is a specific instance of a contextual bandit problem with two possible actions and a reward function that is 1 if the accepted applicant returns the loan, -1 if he doesn't and 0 if the lender decides to reject the applicant. There exist a multitude of methods for this setting ranging from adversarial bandit approaches such as EXP4 Auer et al. (2002) to methods that rely on simple parametric assumptions on the data (for example linear responses) such as OFUL Abbasi-Yadkori et al. (2011) or GLM Filippi et al. (2010). These methods' applicability to the BLP depends on the data satisfying restrictive parametric assumptions. Specifically, these methods are not compatible with the use of deep neural network (DNN) function approximators.

**Neural solutions to the bank loan problem**  More recently, deep neural networks have been successfully applied to this class of problems, with several proposed strategies (Riquelme et al. (2018),Zhou et al. (2020),Xu et al. (2020)). For example, in Riquelme et al. (2018) the authors conduct an extensive empirical evaluation of existing bandit algorithms on public datasets. A more recent line of work has focused on devising ways to add optimism to the predictions of neural network models. Methods such as Neural UCB and Shallow Neural UCB Zhou et al. (2020); Xu et al. (2020) are designed to add an optimistic bonus to the model predictions that is a function of the representation layers of the model. Their theoretical analysis is inspired by insights gained from Neural Tangent Kernel theory. Other recent works in the contextual bandit literature such as Foster & Rakhlin (2020) have started to pose the question of how to extend theoretically valid guarantees into the function approximation scenario, so far with limited success Sen et al. (2021).

Although neural bandit methods such as Neural UCB can be directly applied to the BLP through its reduction to a contextual bandit problem, the current state of the art on these problems is achieved by the PLOT algorithm from Pacchiano et al. (2021). PLOT relies on a data filtering mechanism that is combined with an optimistic labelling strategy to choose which points $\mathbf{x}$ to assign the optimistic pseudo-label. However, this approach suffers from increased false acceptance rate in particular early on in the learning process. In Pacchiano et al. (2021) the authors mitigate this problem by assigning pseudo-labels with fixed (small) probability $\epsilon$, however this approach is rather arbitrary and doesn't take into account the intrinsic properties of the dataset or a candidate.

**Selective labels**  The bank loan problem is also known as the"selective labels" problem (Lakkaraju et al., 2017), but the majority of the literature in this area so far focuses on the offline setup. The recent paper of Wei (2021) considers online scenario, but in contrast with our setting their online mechanism requires the data distribution to be stationary. Additionally although the authors mention their methods can be extended to the setting of NNs, their experimental evaluation is conducted mainly over linear models.

**Reject inference**  This line of methods attempts to correct bias in credit scoring by modelling the missing outcomes of the rejected applications. While statistical and machine learning methods are most commonly used, more recently deep learning methods have also been applied, e.g. Mancisidor et al. (2020). However, like in the majority of selective labels literature, the online scenario has not been considered.

**Fairness** Fairness is a central concern in algorithmic decision-making and existing literature offers numerous approaches and perspectives. Although systematically assessing fairness aspects is not the goal of our present work we feel compelled to mention a selection of papers. Fairness in the contextual bandits setting is explored by e.g. Joseph et al. (2016), Schumann et al. (2019). Domain adaptation techniques for censoring of protected features were proposed and evaluated by Edwards & Storkey and Zhang et al. (2018) among others and Coston et al. (2019), Singh et al. (2021) address fairness in the context of distributional shift. The above works do not consider the online setting, or more specifically the BLP. In the setup of "selective labels" Kilbertus et al. (2020) considers learning under fairness constraints under assumption of stationary data distribution (not required by our method) and Coston et al. (2021) proposes a method for selecting fair models from existing ones.

**Repeated Loss Minimization** Previous work Hashimoto et al. (2018) have studied the problem of repeated classification settings where the underlying distribution is influenced by the model itself. The works Miller et al. (2021); Perdomo et al. (2020) and Hardt et al. (2016) extend this idea to introduce the problem domains of *performative prediction* and *strategic classification*. Unlike in the above setting, in our case only the labelled training set is affected by the model, which results in a distributional shift *between* this set and the set of points presented to the model. This shift is the reason for using *domain adaptation* in our setting.

## 3 BACKGROUND

### 3.1 THE BANK LOAN PROBLEM

We are focusing on the class of sequential learning tasks, where at every timestep $t$, the learner is presented with a query $\mathbf{x}_t \in \mathbb{R}^d$ and needs to predict its binary label $y_t \in \{0, 1\}$. Acceptance of the query carries an unknown reward of $2y_t - 1$ and rejection a fixed reward of $0$. Using the bank loan analogy, the lender receives a unit gain for accepting a customer who repays his loan, experiences a unit loss for accepting a customer who defaults and receives nothing if they reject an application. The learners objective is to maximize their reward, which is equivalent to maximizing the number of correct decisions made during training (Kilbertus et al., 2020).

We assume that the distribution of the labels is given by a probability function $\rho_{\theta_\star}$:

$$y(\mathbf{x_t}) = \begin{cases} 1 & \text{with probability } \rho_{\theta_\star}(\mathbf{x}_t) \\ 0 & \text{o.w.} \end{cases} \tag{1}$$

where $\rho_{\boldsymbol{\theta}_\star} : \mathbb{R}^d \to \mathbb{R}$ is a function parameterized by $\boldsymbol{\theta}_\star \in \Theta$. We assume that the conditional distribution of the responses is fixed for all $t$, although in principal the distribution of the *query points* $\mathcal{P}(\mathbf{x})$ may be different for different $t$'s: e.g. the bank may extend it's loan provision at a certain time point $t_0$ and start targeting a different applicant demographic.

We assume that the learners decision at time $t$ is parameterized by $\boldsymbol{\theta}_t$ and is given by the rule:

$$\text{If } \rho_{\boldsymbol{\theta}_t}(\boldsymbol{x}_t) \geq \frac{1}{2} \text{ accept}$$

A common measure of the learners performance is *regret*:

$$\mathcal{R}(t) = \sum_{\ell=1}^{t} \max(0, 2\rho_{\boldsymbol{\theta}_\star}(\mathbf{x}_\ell) - 1) - a(\mathbf{x}_\ell)(2\rho_{\boldsymbol{\theta}_\star}(\mathbf{x}_\ell) - 1)$$

where we denote by $a(\mathbf{x}_\ell) \in \{0, 1\}$ the indicator of whether the learner has decided to accept (1) or reject (0) the data point $\mathbf{x}_\ell$.

Minimizing regret is a standard objective in the online learning and bandits literature (see Lattimore & Szepesvári (2020)).

Using the set of accepted data points, i.e. the *greedy* approach, $\rho_{\boldsymbol{\theta}_t}$ can be obtained by minimising binary cross entropy loss:

$$\mathcal{L}(\boldsymbol{\theta}|\mathcal{A}_t) = \sum_{\mathbf{x} \in \mathcal{A}_t} -y(\mathbf{x}) \log\left(\rho_{\boldsymbol{\theta}}(\boldsymbol{x})\right) - (1 - y(\mathbf{x})) \log\left(1 - \rho_{\boldsymbol{\theta}}(\mathbf{x})\right) \tag{2}$$

where $\mathcal{A}_t = \{\mathbf{x}_\ell | a_\ell = 1 \text{ and } \ell \leq t - 1\}$ denotes the dataset of accepted points up to the time $t - 1$. In what follows we call such $\rho_{\boldsymbol{\theta}_t}$ the *biased model*.

## 3.2 THE PSEUDO-LABEL OPTIMISM APPROACH TO BLP (PLOT)

Learners that derive their strategy solely from training on accepted data points are prone to acquiring rejection bias, and thus they accumulate infinite regret. However, the $\epsilon$-greedy strategy of randomly accepting candidates risks accepting too many false-positives, leading to the same outcome. In Pacchiano et al. (2021) the authors propose *pseudo-label optimism* (PLOT), the simple yet effective approach to optimism. The PLOT approach can be summarized as follows: to each new query $\mathbf{x}_0$ *rejected* by the greedy model it optimistically assigns a positive *pseudo-label*, retrains the model on the dataset that includes this data point with positive label and uses the resulting *pseudo-label* model to confirm or ignore the optimistic prediction. Formally, PLOT adds a term to the loss function 2:

$$\mathcal{L}^{\mathcal{C}}(\boldsymbol{\theta}|\mathcal{A}_t) = \mathcal{L}^\lambda(\boldsymbol{\theta}|\mathcal{A}_t \cup \{(\mathbf{x}_0, 1)\})$$
$$= \sum_{\mathbf{x} \in \mathcal{A}_t} -y(\mathbf{x}) \log\left(\rho_{\boldsymbol{\theta}}(\mathbf{x})\right) - (1 - y(\mathbf{x})) \log\left(1 - \rho_{\boldsymbol{\theta}}(\mathbf{x})\right) - \underbrace{\log(\rho_{\boldsymbol{\theta}}(\mathbf{x}_0))}_{\text{pseudo-label loss}} \quad (3)$$

## 3.3 ADVERSARIAL DOMAIN ADAPTATION

Domain adaptation studies strategies for learning in the presence of distributional shift between the labelled data used for training and the unlabelled data domain containing test samples. Adversarial domain adaptation, first proposed in Ganin et al. (2016), is based on the theory of distance between domains introduced in Ben-David et al. (2006; 2010) and the GAN approach of Goodfellow et al. (2014). The main idea of this method is constructing a representation of the data that is both discriminative for the given classification task and domain invariant. Further work in adversarial domain adaptation has generalised the method to different adversarial loss functions and generative tasks, e.g. (Liu & Tuzel, 2016; Tzeng et al., 2017; Zhang et al., 2018), and successfully applied it to a wide array of challenges, e.g. (Wang et al., 2019; Edwards & Storkey). Our approach in the present work is similar to the DANN algorithm of Ganin et al. (2016), that we found to give stable results for the duration of the training.

Given a labelled *source* dataset $\mathcal{S} \subset \mathbb{R}^d$ and a *target* dataset $\mathcal{T} \subset \mathbb{R}^d$ whose labels we need to predict, the adversarial domain adaptation approach is to simultaneously train a generator $G_\theta : \mathbb{R}^d \to \mathbb{R}^{d'}$ that encodes the data, a classifier $C_\psi : \mathbb{R}^{d'} \to \{0, 1\}$, and a discriminator $D_\phi : \mathbb{R}^{d'} \to \{0, 1\}$ that discriminates between the encodings of samples from $\mathcal{S}$ and $\mathcal{T}$, so that $C_\psi(G_\theta(\cdot))$ has a high classification accuracy on $\mathcal{S}$, and $G_\theta$ is trained adversarially with $D_\phi$ to minimise the possibility of distinguishing between samples from $G_\theta(\mathcal{S})$ and $G_\theta(\mathcal{T})$. Following Zhang et al. (2018) we will refer to the representation of the data by $G_\theta$ as "de-biased" and to $C_\psi$ as the "de-biased classifier" - "de-biased" in our context refers to a representation that doesn't distinguish between test and train domains.

The training of generator, classifier, and discriminator can be summarised by the training objective:

$$\min_\theta \min_\psi \max_\phi \mathbb{E}_{\mathbf{x} \in \mathcal{T}} \log(D_\phi(G_\theta(\mathbf{x})) + \mathbb{E}_{\mathbf{x} \in \mathcal{S}} \log(1 - D_\phi(G_\theta(\mathbf{x}))) + L^C_{x \in \mathcal{S}}(C_\psi(G_\theta(\mathbf{x})), y(\mathbf{x})), \quad (4)$$

where $L^C$ is the binary cross entropy loss and $y : \mathcal{S} \to \{0, 1\}$ provides the labels.

## 4 ADVERSARIAL OPTIMISM

In this section we describe our main contribution, Adversarial Optimism (AdOpt) in detail. AdOpt uses two classifiers. The first one is a "biased" classifier trained on *all the accepted data* thus far. The second one is our adversarially de-biased classifier. AdOpt then proceeds as follows:

- If a data point is *accepted* by the biased classifier, we accept it and add it to the dataset with the true label.
- If a data point is instead *rejected* by the biased classifier, we use de-biased classifier to decide whether to add it to the pseudo-label dataset.

- As in PLOT, retrain on the pseudo-label candidates with *optimistic labels* to decide final acceptance.

### 4.1 ADVERSARIAL DE-BIASING IN THE BLP CONTEXT

Let us recap the task at hand: in the BLP setting 3.1 at time $t$ we possess a labelled set of previously accepted applicants $\mathcal{A}_t$ and we want to utilize this information to label a new applicant. The obstacle to overcome is that the distribution of the data points in the general applicant pool $\mathcal{A}$, is different from the distribution of data points in $\mathcal{A}_t$, that was influenced by the previous choices of the classifier. This will compromise the performance of a classifier trained on $\mathcal{A}_t$. We address this issue by using the domain adaptation method of Section 3.3 to find the de-biased representation $G_\theta$ of the data in $\mathcal{A}_t$ and train the de-biased classifier $C_\phi$ to predict a label for the candidate $\mathbf{x}$ sampled from $\mathcal{A}$.

We expect $C_\phi$ to give *optimistic* predictions for the labels of $\mathbf{x} \in \mathcal{A}$, in the following sense: as shown by Edwards & Storkey, adversarial domain adaptation produces a classifier and generator that converge to approximating the *demographic parity* property

$$P(C_\psi(G_\theta(\mathbf{x})) = 1 \mid \mathbf{x} \in \mathcal{S}) = P(C_\psi(G_\theta(\mathbf{x})) = 1 \mid \mathbf{x} \in \mathcal{T}). \tag{5}$$

Equation 5 states that the percentage of positive predictions on the transformed labelled set $G_\theta(\mathcal{S})$ is the same as on $G_\theta(\mathcal{T})$. In the BLP setting, as the training progresses and the classifier becomes better at accepting the right candidates, the distribution of true positive labels in $\mathcal{A}_t$ diverges from that of the general population: for $t \gg 0$

$$P(y(\mathbf{x}) = 1 | \mathbf{x} \in \mathcal{A}_t) > P(y(\mathbf{x}) = 1 | \mathbf{x} \in \mathcal{A}). \tag{6}$$

Note that the above is simply a consequence of the classifier being successful in its job and is independent of the accumulation of bias due to erroneous rejections. Since in the adversarial adaptation approach the classifier $C_\psi$ is trained to preserve the accuracy on the labelled set $\mathcal{A}_t$, $P(C_\psi(G_\theta(\mathbf{x})) = 1)$ is going to be close to $P(y_{true}(\mathbf{x}) = 1 | \mathbf{x} \in \mathcal{A}_t)$ *both* when $\mathbf{x} \in \mathcal{A}_t$ *and* $\mathbf{x} \in \mathcal{A}$. In the BLP setting, the adversarial training regime creates de-biased classifier that optimistically overpredicts positive labels for the incoming queries.

To verify empirically that our adversarial training regime in the context of BLP produces de-biased classifier with properties as above, we create a performance metrics log across the experiments comparing the performance of the de-biased classifier and the classifier trained on the original biased data. The mean and standard deviation for the de-biased classifier that is trained from scratch at each step are in Table 1. The last two columns empirically evaluate Equation 5.

As can be seen from Table 1, increased recall of the adversarially de-biased classifier comes at the expense of lower precision. This trade off can be regulated by the length of adversarial training. We empirically found that for minimizing regret best performance is achieved when training for small number of epochs at each step without resetting the weights of the adversarial triad. This approach also reduces the training duration.

### 4.2 THE ADOPT METHOD

A known drawback of the adversarial training is its instability. It precludes the standalone adversarial classifier from reliably achieving the optimal results. Our experiments show that the standalone adversarial classifier is highly sensitive to the number of training epochs, hence hyperparameter tuning is required for optimal performance. To mitigate this, we combine the adversarially de-biased classifier with the PLOT approach as illustrated in Figure 1. If the point is rejected by the biased classifier and recommended for acceptance by the de-biased classifier we evaluate this recommendation against the models prior knowledge by adding it to the original dataset with positive pseudo-label and retraining an optimistic classifier to make a final decision. Incorporating prior model knowledge via PLOT helps to further reduce regret incurred by our algorithm.

Using the de-biased classifier for assigning pseudo-labels enables AdOpt to explore faster. As explained in 4.1, the de-biased classifier possesses increased recall. While this comes at expense of lower precision, its acceptance suggestions are superior to randomly picking pseudo-label candidates as is done in Pacchiano et al. (2021). Since lowering the threshold in the biased classifier also has the effect of trading precision and recall we also compare AdOpt with pseudo-labelling by

Table 1: Mean and STD values for performance metrics of biased and de-biased classifiers on the "Adult" dataset. We find that adversarial de-biasing trades off precision and recall. All columns except the last contain metrics for the incoming batch, or "Target dataset" $\mathcal{T}$. The last column contains the proportion of predicted positives for the "Source dataset" $\mathcal{S}$ to illustrate equation 5.

| | | Recall | | Precision | | Predicted Positive | | |
|---|---|---|---|---|---|---|---|---|
| | True Pos. | Bias | No Bias | Bias | No Bias | Bias | No Bias | No Bias $\mathcal{S}$ |
| Mean | 0.24 | 0.59 | 0.83 | 0.78 | 0.31 | 0.18 | 0.66 | 0.69 |
| STD | 0.07 | 0.2 | 0.16 | 0.19 | 0.12 | 0.07 | 0.11 | 0.11 |

changing threshold of the biased classifier in Figure 6. AdOpt shows superior performance to both approaches with minimal hyperparameter tuning.

During the exposition in 3, we considered the setting where only one point is acted upon in each time-step. In practice however it is more computationally effective to consider the scenario, where at each time-step a learner receives *a batch* of data points $\mathcal{B}_t = \{\mathbf{x}_{t,1}, \ldots, \mathbf{x}_{t,B}\}$, and sees labels for accepted points $(y_{t,1} \ldots, y_{t,B'}), y_t \in \{0,1\}$, where $B$ is the size of the batch, and $B'$ is the number of accepted points. All the notions from 3.1 and 3.2 extend naturally to this setting. We train a de-biased classifier adversarially for every timestep on the pair $(\mathcal{A}_t, \mathcal{B}_t)$ and use it to obtain optimistic predictions for the labels of the data points in $\mathcal{B}_t$. We then make use of the pseudo-label filtering from 3.2, where the loss 2 is adjusted to be computed over the batch:

$$\mathcal{L}^{\mathcal{C}}(\boldsymbol{\theta}|\mathcal{A}_t, \mathcal{B}_t) = \mathcal{L}(\boldsymbol{\theta}|\mathcal{A}_t \cup \{(\mathbf{x}, 1) \text{ for } \mathbf{x} \in \mathcal{B}_t\})$$
$$= \sum_{\mathbf{x} \in \mathcal{A}_t} -y \log\left(\rho_{\boldsymbol{\theta}}(\mathbf{x})\right) - (1-y)\log(1 - \rho_{\boldsymbol{\theta}}(\mathbf{x})) - \underbrace{\sum_{\boldsymbol{x} \in \mathcal{B}_t^0} \log(\rho_{\boldsymbol{\theta}}(\mathbf{x}))}_{\text{pseudo-label loss}}, \quad (7)$$

where $\mathcal{B}_t^0 \subset \mathcal{B}_t$ denotes the subset of points recommended for acceptance by the de-biased classifier.

---

**Algorithm 1** AdOpt
---
Initialize accepted dataset $\mathcal{A}_1 = \emptyset$
**for** $t = 1, \cdots, T$ **do**

> 1. Observe batch $\mathcal{B}_t = \{\mathbf{x}_t^{(j)}\}_{j=1}^B$.
> 2. Run adversarial domain adaptation algorithm 2 with starting weights initialized from training $\mathcal{B}_{t-1}$ on $(\mathcal{A}_t, \mathcal{B}_t)$ to obtain generator and classifier pair $(G, C)$.
> 3. Train the biased model $\rho_{\boldsymbol{\theta}_t}$ by minimizing the loss $\mathcal{L}_t(\boldsymbol{\theta}|\mathcal{A}_t)$.
> 4. Compute the pseudo-label filtered batch $\widetilde{\mathcal{B}}_t = \{(\mathbf{x}_t^{(j)}, 1) \mid \rho_{\boldsymbol{\theta}_t}(\mathbf{x}_t^{(j)}) < \frac{1}{2} \text{ and } C(G(x_t^j)) = 1\}$.
> 5. Train the pseudo-label model by minimizing the optimistic pseudo-label loss, $\mathcal{L}_t^{\mathcal{C}}(\boldsymbol{\theta}|\mathcal{A}_t, \widetilde{\mathcal{B}}_t)$
> 6. For $\mathbf{x}^{(j)} \in \mathcal{B}_t$ compute acceptance decision via $a_t^{(j)} = \begin{cases} 1 & \text{if } \rho_{\boldsymbol{\theta}_t^C}(\boldsymbol{x}_t) \geq \frac{1}{2} \text{ or } \rho_{\boldsymbol{\theta}_t}(\boldsymbol{x}_t) \geq \frac{1}{2} \\ 0 & \text{o.w.} \end{cases}$
> 7. Update $\mathcal{A}_{t+1} \leftarrow \mathcal{A}_t \cup \{(\mathbf{x}_t^{(j)})\}_{j \in \{1, \cdots, B\} \text{ s.t. } a_t^{(j)}=1}$.

**end**

---

During the very first time-step ($t = 1$), the algorithm accepts all the points in $\mathcal{B}_1$. In subsequent time-steps the adversarial domain adaptation algorithm is trained on the pair $(\mathcal{A}_t, \mathcal{B}_t)$ to produce the pair consisting of de-biased classifier and the generator of de-biased data representation $(C, G)$. Next we identify a subset of the current batch composed of those points that are both currently being predicted as rejects by the biased model and have been selected as potential positives by the de-biased classifier. They comprise the pseudo-label batch of potential candidates for acceptance. We train the pseudo-label model on the combined dataset and accept those data points, whose labels were confirmed by the pseudo-label model predictions.

A notable feature of PLOT is that as the size of the dataset of accepted points grows, the predictions of the pseudo-label model differ less and less from the predictions of the biased model trained on the set of the accepted points. Ideologically once the dataset is sufficiently large and accurate information can be inferred about the true labels, the inclusion of $\widetilde{\mathcal{B}}_t$ into the pseudo-label loss has vanishing effect on the model's predictions. The latter has the beneficial effect of making false positive rate decrease with $t$. This balances the opposite trend in the predictions of the de-biased adversarial classifier, that tends to predicting more positives as the training progresses as a result of the Equations 6 and 5.

We provide the details of the adversarial training algorithm in the Appendix A.1

## 5 EXPERIMENTS

**Datasets and methods**   To assess the performance of AdOpt, we consider five benchmark datasets: the UCI datasets *Adult*, *Portuguese Bank*, *Communities and Crime*, and *German Credit Data*  Lichman et al. (2013) as well as MNIST converted to the binary reward format of the Bank Loan problem by defining the positive class to be images of the digit 5. If a point is accepted, it receives reward 1 if its true label is positive, and -1 otherwise. If a point is rejected, reward is always 0.

We compare against four baseline methods: PLOT, NeuralUCB, Greedy (no exploration), and decayed $\epsilon$-greedy method. For $\epsilon$-greedy, we follow (Kveton et al., 2019) and use a decayed schedule, dropping to 0.1% exploration by T=2500. We combine PLOT with $\epsilon$-greedy selection of pseudo-label candidates as in Pacchiano et al. (2021). In addition we evaluate the performance of the standalone adversarially de-biased classifier as an ablation study.

Since sampling of batches presented during training as well as stochastic nature of optimisation has a significant effect on the final regret, to obtain the results in this paper we average the performance of each method across 5 different batch sampling methods and 5 separate runs of 2500 timesteps each, so altogether 25 experiments for each method. We note that this is different from the approach in Pacchiano et al. (2021) where only one specific way of sampling examples from the datasets was examined. This explains the difference in our evaluation of performance of PLOT on Bank and MNIST where it tied with NeuralUCB (as evidenced by average t-values of respectively $-1.4792$ and $0.4754$ obtained in our experiments). We also note that, as opposed to Pacchiano et al. (2021), we report absolute values of regret at every step, rather than regret with respect to a baseline model, which allows us to achieve more transparent results compared to Pacchiano et al. (2021). The high number of experiments allows us to conduct t-test to confirm statistical significance of our results, with t-values of $> 2$ indicating less than 5 percent chance of null-hypothesis.

The number of training epochs for the adversarially de-biased classifier is a parameter that can in principle be tuned to optimise the results for each dataset. The number of epochs also influences the runtime of the AdOpt algorithm. For the shortest possible duration of one epoch per batch the runtime is comparable to that of PLOT. However compute has relatively low priority for our purposes here: compute is cheap while mistakes are costly. We illustrate the effect of increasing the number of epochs per batch on *Crime* and *German* datasets in Figures  3 and  4. Further particulars on experiment details and hyperparameters can be found in Appendix  A.2

**Main results**   We report the following main results. First, in Table  1, we compare the performance metrics for the adversarially de-biased classifier (trained with resetting of weights for 30 epochs for every batch) and the greedy classifier trained on the original biased data in a training run on the *Adult* dataset. The de-biased classifier achieves higher recall while maintaining predictive ability evidenced by its precision value. Our code generates a full log of performance metrics for the biased and de-biased classifiers for every run of the algorithm.

Second, in Figures  2 and  3,  4 , we report cumulative regret for each method. AdOpt outperforms all other methods in our experiments by a significant margin on *Adult* and MNIST even when the triad of generator, discriminator and classifier is trained for 1 epoch only for every batch (without resetting the weights). The respective t-values are 7.6778 and 3.8547. When training with 1 epoch AdOpt ties with PLOT on *Bank*, *Crime* and *German* datasets with t-values of $-0.7147$, $0.3842$ and $-0.6844$. However increasing the length of adversarial training at each batch improves AdOpt performance on these datasets (see Figures  3 and  4). Specifically with training for 10 epochs at each batch

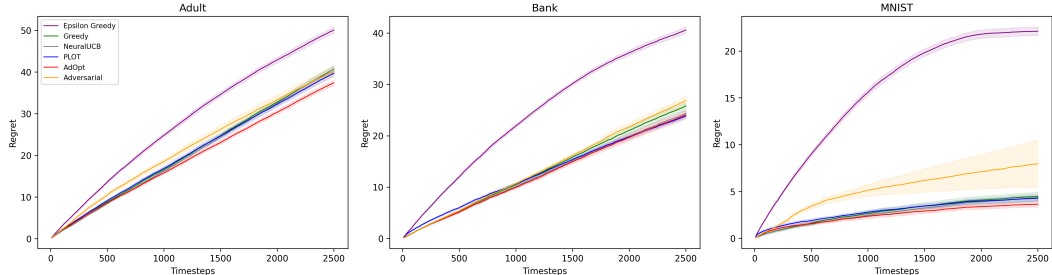

Figure 2: Cumulative regret of AdOpt vs. PLOT, NeuralUCB, Greedy, Epsilon greedy and Standalone Adversarial classifier. The adversarially de-biased component of AdOpt is trained for 1 epoch at each batch. Cumulative regret is reported as a function of timestep, for 2500 timesteps. The shaded region represents one standard deviation from the mean, computed across 25 experiments, 5 runs over 5 different sampling methods. We find that AdOpt outperforms other methods by a large margin on *Adult* and MNIST (t-values of 7.6778 and 3.8547 at 2500 steps) and matches PLOT and NeuralUCB on *Bank*(respective t-values of $-0.7147, 0.6602$). t-values of $> 2$ indicate less than 5 percent chance of null-hypothesis

AdOpt shows statistically significant advantage over PLOT on *Crime* (t-value of $2.4451$) and gains advantage over PLOT on *German* with t-value of $1.6827$ which indicates a trend of approaching statistical significance.

We also evaluate and plot the performance of the standalone adversarially de-biased classifier trained for the same number of epochs as for AdOpt at every batch a an ablation study.

Third, our initial evaluation of bias with respect to gender on the Adult dataset shown in Figure 5 in the Appendix indicates that the adversarially de-biased classifier exhibits significantly less bias in its predictions for different subgroups within those categories than the other methods that we evaluated. Thus it significantly reduces the discrimination of candidates with various characteristics within those categories, even though it was not specifically trained to be unbiased with respect to them. We plan to further investigate this encouraging observation in the sequel to present work.

## 6 CONCLUSION

We apply adversarial domain adaptation to online learning in the Bank Loan Problem setting to address the distribution shift between the labelled training dataset and the true distribution present due to past acceptance or rejection decisions. We present AdOpt, a novel algorithm that experimentally establishes a new state-of-the-art across a variety of instances of the BLP, significantly outperforming current methods, even with minimal hyperparameter tuning.

In general, we expect the performance of domain adaptation-based methods like AdOpt to have the biggest impact on datasets with difficult exploration problems. In future work we see three key steps towards leveraging domain adaptation for increasingly difficult exploration problems:

- Harder datasets for the bank loan problem, which can include non-stationary or adversarial datasets.
- Generalization of AdOpt to higher dimensional actions, enabling application to new, more complex problem settings.
- Supported by work on the first two items, consideration of the general Reinforcement Learning regime, with sequential states and actions.

Our preliminary experimental results indicate that the adversarially de-biased classifier has the potential to greatly reduce the risk of *self-reinforcing rejection-loops* for the class of decision making problems that we considered. This might result in fairer access to bank loans for underrepresented minorities and overall more unbiased algorithmic decision making. This finding makes adversarial-based methods promising candidates for further exploration in this direction.

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

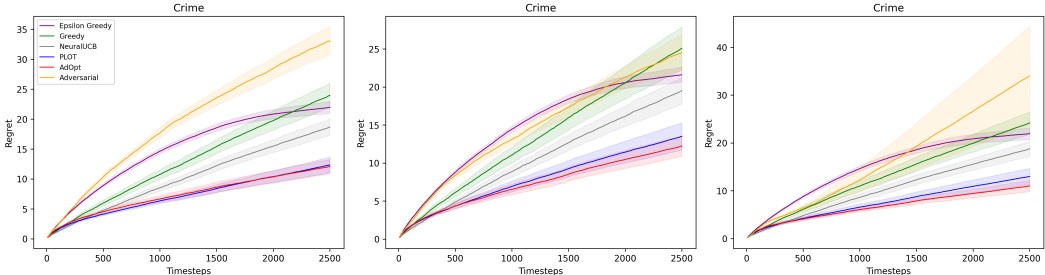

Figure 3: Cumulative regret of AdOpt vs. PLOT, NeuralUCB, Greedy, Epsilon greedy and Standalone Adversarial classifier on the *Crime* datasets when training adversarially de-biased classifier for 1, 2 and 10 epochs at every batch. Cumulative regret is reported as a function of timestep, for 2500 timesteps. The shaded region represents one standard deviation from the mean, computed across 25 experiments. We find that increasing the length of adversarial training at each batch improves the performance of AdOpt. At 10 batches per epochs AdOpt outperforms the next leading method, PLOT with t-value of 2.4451. t-values of $> 2$ indicating less than 5 percent chance of null-hypothesis

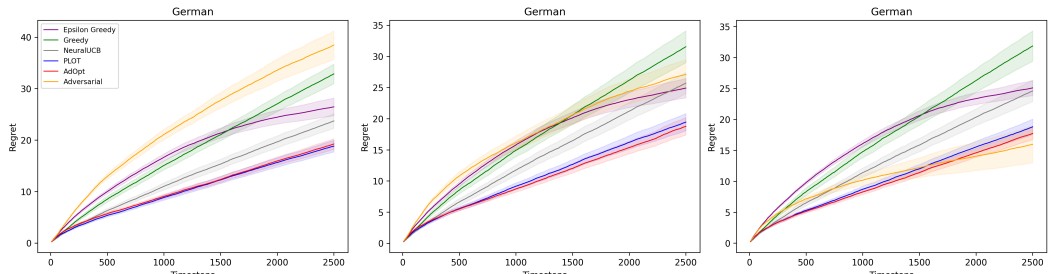

Figure 4: Cumulative regret of AdOpt vs. PLOT, NeuralUCB, Greedy, Epsilon greedy and Standalone Adversarial classifier on the *Crime* datasets when training adversarially de-biased classifier for 1, 2 and 10 epochs at every batch. Cumulative regret is reported as a function of timestep, for 2500 timesteps. The shaded region represents one standard deviation from the mean, computed across 25 experiments. We find that increasing the length of adversarial training at each batch improves the performance of AdOpt. At 10 batches per epochs AdOpt outperforms the next leading method, PLOT with t-value of 1.6826. t-values of $> 2$ indicating less than 5 percent chance of null-hypothesis

# A  APPENDIX

## A.1  ADVERSARIAL TRAINING ALGORITHM

Given the datasets $\mathcal{S}$ and $\mathcal{T}$ of labelled and unlabelled data the algorithm 2 trains a classifier, generator and discriminator triad as in 3.3. Before beginning of the optimisation, $\mathcal{S}$ and $\mathcal{T}$ are augmented by adding a $y$-label to each data point. In $\mathcal{S}$ the value of the $y$-label is the known true label for each data point and in $\mathcal{T}$ we set the $y$-label to be zero (the values of $y$ on $\mathcal{T}$ are not used for optimization). We also add a $z-$label to all the data points indicating whether the data point originated from $\mathcal{S}$ or $\mathcal{T}$.

At each step of the optimisation, a batch of size $B$ is sampled from the union of the augmented datasets. The classifiers predicted $y-$label and discriminators predicted $z-$label are computed for each element in the batch. If there are samples that originated from $\mathcal{S}$, then the binary cross entropy loss of the predicted $y-$label is calculated for those elements, else it is set to zero. The negative binary cross entropy loss of the predicted $z-$label is calculated for all the elements of the batch. These losses are then combined, using the hyperparameter $\lambda$ to control the relative weighting of the classifier and discriminator's loss. Finally, the gradient of the combined loss is used to update the

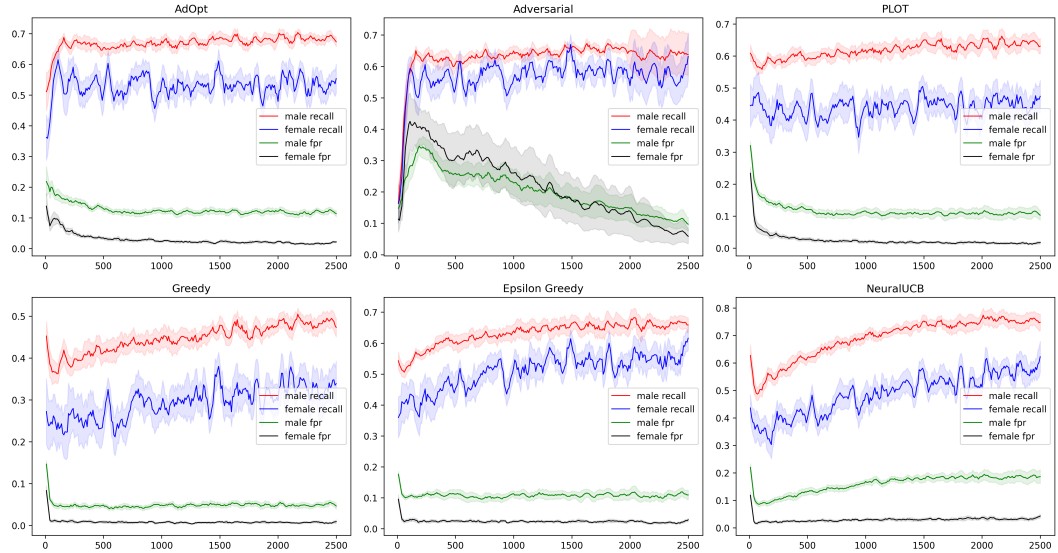

Figure 5: The recall and false positives ratio (fpr) values per batch for Females and Males subgroups of the Adult dataset for the different methods evaluated in the paper. The adversarially de-biased classifier is the only one among evaluated methods that shows similar values of recall and fpr for these groups, thus coming close to satisfying the equality of odds definition of fairness. This remarkable property of the adversarial classifier is a potentially very interesting subject of future study.

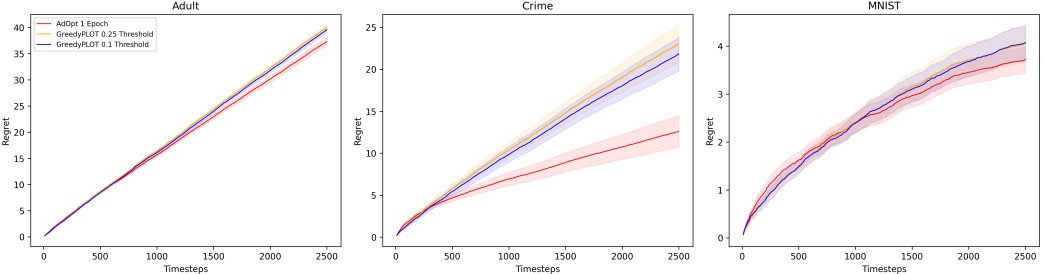

Figure 6: Cumulative regret of AdOpt vs PLOT combined with pseudo-labelling using biased classifier at thresholds $0.1$ and $0.25$ ("GreedyPLOT"). While lowering the threshold of the biased classifier also has an effect of trading recall and precision, adversarial domain adaptation is more successful at finding the optimal threshold for minimizing cumulative regret and it does so with no hyperparameter tuning.

---

**Algorithm 2** Adversarial Domain Adaptation

---

**Input:** Unlabelled data $\mathcal{T} = \{\mathbf{x}^{(j)}\}_{j=1}^{N_{\mathcal{T}}}$, labelled data $\mathcal{S} = \{(\mathbf{x}^{(j)})\}_{j=1}^{N_{\mathcal{S}}}$, batch size $B$, number of batches $N_B$, and adversarial weight $\lambda$.
**Output:** The generator $G_\theta$ and classifier $C_\psi$.

Let $\hat{\mathcal{T}} = \{(\mathbf{x}^{(j)}, 0, 1) \mid \mathbf{x}^{(j)} \in \mathcal{T}\}$ and $\hat{\mathcal{S}} = \{(\mathbf{x}^{(j)}, y(\mathbf{x}^{(j)}), 0) \mid \mathbf{x}^{(j)} \in \mathcal{S}\}$.
Initialise the generator $G_\theta$, classifier $C_\psi$, and discriminator $D_\phi$ with parameters $\theta$, $\psi$, and $\phi$ respectively.
**for** $n = 1, \cdots, N_B$ **do**
    1. Sample batch $\{(\mathbf{x}_i, y_i, z_i)\}_{i=1}^B$ from $\hat{\mathcal{T}} \cup \hat{\mathcal{S}}$.
    2. Compute
$$\hat{Y} = \{\hat{y}_i \mid \hat{y}_i = C_\psi(G_\theta(\mathbf{x}_i))\}_{i=1}^B$$
    and
$$\hat{Z} = \{\hat{z}_i \mid \hat{z}_i = D_\phi(G_\theta(\mathbf{x}_i))\}_{i=1}^B.$$
    3. Compute $B' = \sum_{i=1}^{B_S}(1 - z_i)$.
    4. **if** $B' = 0$ **then**
$$L_C = 0$$
    **else**
$$L_C = -\frac{1}{B'}\sum_{i=1}^B (1 - z_i)[y_i \log(\hat{y}_i) + (1 - y_i)\log(1 - \hat{y}_i)],$$
    **end**
    5. Compute the discriminator loss,
$$L_D = \frac{1}{B}\sum_{i=1}^B z_i \log(\hat{z}_i) + (1 - z_i)\log(1 - \hat{z}_i).$$
    6. Compute the combined loss, $L = L_C + \lambda L_D$.
    7. Update $\theta$, $\psi$, and $\phi$ using an optimiser and $\nabla_\theta L$, $\nabla_\psi L$, and $-\nabla_\phi L$ respectively.
**end**

---

generator and classifier's parameters, and the negative of the gradient of the combined loss is used to update the discriminator's parameters.

## A.2 EXPERIMENT DETAILS AND HYPERPARAMETERS

Our experiments use the optimal hyperparameters for each method reported in previous papers. NeuralUCB uses an alpha of $0.4$, and a discount factor of $0.9$. PLOT uses a two-layer, 40-node, multi-layer perceptron. AdOpt is trained using an encoded dimension of 100, as well as a hidden layer size of 100. Lambda is set to 1, and we train the adversarial model with Binary Cross Entropy loss. Each experiment runs on a single GPU, and our results were collected by trivially distributing 25 experiments in parallel on 25 different GPU cores.

For the adversarial training, the generator, classifier, and discriminator are multi-layer perceptrons, where the dimension of the input, hidden layer(s), and output are [Data Dimension, 100, 100, 100], $[100, 100, 1]$, and $[100, 100, 100, 1]$ respectively. We used Adam optimizer, with $(\beta_1, \beta_2) = (0, 0.99)$ and a learning rate of $0.0005$ for the discriminator and $0.0001$ for the generator and classifier Kingma & Ba (2014).

As the historical dataset of accepted points grows, there is a large disparity in size between the sample (new batch) and target (historical) datasets for adversarial domain adaptation. To handle this data size discrepancy, we restrict the target dataset for adversarial training to a randomly sampled

batch of size $32 * 100$ from the historical data, once the data reaches this size. As we use a batch size of 32, this occurs after 100 time steps.

