# OpenReview forum: "Unbiased Decisions Reduce Regret: Adversarial Optimism for the Bank Loan Problem"
_ICLR.cc/2023/Conference — Submitted to ICLR 2023_

### Official Review · Reviewer_cz4a · 2022-10-22

**Confidence:** 3
**Correctness:** 3
**Technical Novelty And Significance:** 3
**Empirical Novelty And Significance:** 1
**Recommendation:** 5

**Clarity, Quality, Novelty And Reproducibility:**

As noted above, I think the writing could use some work to be more clear. The approach is a modification of an existing approach, PLOT, but uses a novel mechanism to decide which rejected samples will be added to the dataset with an optimistic pseudolabel. The adversarial domain adaptation approach used to train the classifier used by this mechanism is also borrowed from existing works, though hasn't been used in this setting before. The works seems sufficiently reproducible, assuming the provided code is released.

**Strength And Weaknesses:**

**Strengths** The core idea seems like a reasonable way to to address a shortcoming of an existing method

**Weaknesses** I found the paper hard to follow in general. It felt like the relevant background explaining PLOT and where the domain adaptation technique fits into the picture was scattered around the paper in an unintuitive way;  I had to read some of the sections several times and out of order to understand what was being proposed. Essentially, the difference between the proposed algorithm (AdaOpt) and PLOT is just that AdaOpt uses a different mechanism to decide whether to include a rejected sample in the dataset.

The idea of using a high-recall classifier to decide which examples to optimistically add to the dataset to help mitigate high false acceptance rate of PLOT seems reasonable. However, as far as I can tell there are no concrete performance guarantees here, so the only way we can evaluate the proposed approach is via the experimental results. Unfortunately, I don't think that the results manage to demonstrate the claims of the paper. Since AdaOpt is a modification of PLOT, the main baseline to outperform is PLOT, yet Figure 2 implies that there is no statistically significant difference between PLOT and AdaOpt on any of the datasets except for "Adult", wherein AdaOpt finishes with slightly lower regret in the end.

**Misc** I think Perdomo et. al. (2020) is an relevant citation to discuss in this paper, as it studies risk minimization in settings where the decision-making policy effects the training data distribution, as it does in the bank loan problem.


References
---
- Perdomo, J., Zrnic, T., Mendler-D\"unner, Celestine, & Hardt, M. (2020).
  Performative Prediction. In H. D. III, & A. Singh, Proceedings of the 37th
  International Conference on Machine Learning (pp. 7599–7609). : PMLR.


**Summary Of The Paper:**

This paper studies the bank-loan problem: a binary classification setting where the learner only receives feedback when it assigns an example the positive label. In this setting, training on a dataset consisting only of the accepted examples naturally leads to bias, since the learner has no ability to correct mislabelled rejections. A recent work addresses this by adding rejected samples to the dataset with an optimistic (positive) label. This naturally leads to a higher false acceptance rate. To help alleviate this issue, this paper proposes modifying this approach by using a second classifier with high recall to decide which points to add to the dataset, so that fewer false acceptances make their way into the dataset. This high-recall classifier is trained using an existing adversarial domain adaptation technique.

**Summary Of The Review:**

Overall the paper proposes a reasonable solution to an issue with a state-of-the-art method for this setting, but fails to provide convincing evidence of improvement over the algorithm it is based on (in theory nor experimentally), and so do not recommend this paper for acceptance.

---

> ### Author Response · Authors · 2022-11-10
> **Response to reviewer cz4a**
>
> Thanks for reviewing our paper.
>
> We are sorry to hear that you found the paper hard to follow in places. We have made an effort to thoroughly explain our experimental results here and in the revision draft, and would be thankful if you could read through them once again, since we believe that their assessment in your review is incorrect.
>
> You wrote: *Since AdaOpt is a modification of PLOT, the main baseline to outperform is PLOT, yet Figure 2 implies that there is no statistically significant difference between PLOT and AdaOpt on any of the datasets except for "Adult", wherein AdaOpt finishes with slightly lower regret in the end.*
>
> In the paper by Pacchiano et al the PLOT method was tested on three datasets: MNIST, Adult and Bank. Their experiments were averaged over 5 runs, but every run examined the same specific way of sampling examples from the datasets. We found that the way examples are presented to the algorithm has a significant effect on the final regret, and hence in the experiments in our paper we averaged results over 5 different seeds, 5 runs for each. **Importantly conducting high number of experiments across several sampling methods allows us to use t-test to confirm the statistical significance of our results with high accuracy.** We have also run experiments on two additional datasets: German and Crime. There was a small mistake in t-values we originally reported in the paper - they were averaged over just one way of sampling the dataset, i.e. for one specific seed instead of 5. Since in contrast to that the regret plots in our paper are created by averaging over several sampling strategies, this had caused a mismatch in reporting the results for some datasets. We have corrected the t-values values in the revised draft (Section 5, note also that the regret plots are now in Figures 2, 3 and 4.)
>
> Figure 2 in the paper clearly shows that AdOpt is the leading method showing **significant performance gap on two datasets, Adult and MNIST** with t-values of 7.6778 and 3.8547 respectively, signifying a very high degree of confidence in statistical significance of these results. Generally for 25 experiments we considered t-values of >2 as a cut-off since it rejects the null hypothesis with 95 percent accuracy. In comparison the t-values for PLOT vs NeuralUCB on these datasets are respectively (3.8313 and -1.4792). PLOT, NeuralUCB and AdOpt tie on Bank (PLOT vs AdOpt: -0.7147, NeuralUCB vs AdOpt 0.6602) and PLOT and AdOpt tie on Crime and German with t-values of 0.3842 and -0.6844.(by "tie" we mean here the lower than threshold statistical significance as evidenced by t-values. Negative t-values are used where AdOpt exhibited higher average regret.)
>
> The reported results are for AdOpt running with the shortest possible duration of adversarial training of just 1 epoch for every batch. At this setting the runtime for AdOpt doesn't differ significantly from the runtime PLOT(e.g. running times were respectively 3379.1 sec and 4581.2 for PLOT and AdOpt on Adult, 3415.8 sec 4235.8 sec on MNIST, and similarly for other datasets). However compute has low priority our purpose in this work: compute is cheap while mistakes are costly. By increasing the length of adversarial training at each batch we were able to further improve AdOpt performance. With training for 10 epochs at each batch **AdOpt shows statistically significant advantage over PLOT on Crime (t-value of 2.4451) and it's performance is now better than that of PLOT on German with t-value indicating a trend towards approaching statistical significance (t-value of 1.6827 ).** We have included the exploration of changes in performance of AdOpt depending on the length of adversarial training in the appendix of our revised draft.
>
> Altogether the above results confirm that AdOpt shows superior performance to the other methods we tested. We hope the above analysis changed your assessment of our results. In particular I hope we managed to demonstrate that our empirical evaluations were precise, extensive, and beyond what exists in the literature. We are happy to answer any further questions you might have regarding them or anything else in the paper.

---

> > ### Author Response · Authors · 2022-11-10
> > **Continuation of Response to reviewer cz4a**
> >
> > You wrote: *Essentially, the difference between the proposed algorithm (AdaOpt) and PLOT is just that AdaOpt uses a different mechanism to decide whether to include a rejected sample in the dataset.*
> > The use of adversarial domain adaptation in the online setting is our main contribution and in fact the main focus of our paper. We use PLOT as strategy for incorporating optimism into a DNN model using pseudo-labels. As we explain in the Introduction, for achieving SOTA performance in terms of regret PLOT has to be combined with an additional strategy for choosing pseudo-label candidates. Simply assigning optimistic pseudo-labels to all candidates in the batch leads to unacceptably high final regret. Therefore Pacchiano et al assign pseudo-labels to a fixed proportion of 0.1 random examples.
> > However this choice is clearly arbitrary. Its two major drawbacks are that it is not dependent on either dataset or a sample that is being analyzed. Such a one-size-fits-all approach cannot work and in fact by carefully re-running and analyzing the experiments in the PLOT paper over several dataset sampling strategies in our paper we show that combined with this strategy PLOT fails to show statistically significant improvement in performance over NeuralUCB on two out of three datasets tested in Pacchiano et al (i.e. on MNIST and Bank the corresponding t-values are -1.4792 and 0.4754 respectively). See also Figure 2 in our paper that includes comparison of PLOT against these methods.
> > Our method, AdOpt, offers a data driven alternative to the above selection of pseudo-label candidates, by using adversarial domain adaptation in an online setting. By combining domain adaptation with the pseudo-label approach of PLOT we are able to achieve statistically significant improvement in performance on several benchmark datasets.
> >
> > While the other two reviewers found the exposition clear, and we have invested a lot of effort to give a through explanation of PLOT and its connection to our method in the paper (see especially page 2 of the Introduction, including Figure 1 which explains AdOpt algorithm, as well as section 4 that gives further details) we would be more than happy to implement any further additional concrete suggestions you might have on improving it.
> >
> > Lastly, thank you for pointing us to the paper by Perdomo et al. It studies optimal strategies for predicting *for* the samples drawn from distribution influenced by the model itself, whereas in our case only our labelled training set is affected by the model, and hence we have to deal with the distributional shift *between* the train and test sets (and so have use domain adaptation). However we agree that this circle of ideas can be useful eg for the future applications in this domain and have cited it in the revised draft.

---

> > ### Comment · Reviewer_cz4a · 2022-11-19
> > **Response to the response**
> >
> > > Importantly conducting high number of experiments across several sampling methods allows us to use t-test to confirm the statistical significance of our results with high accuracy
> >
> > I am not particularly familiar with all of these datasets, but I am a bit skeptical that 5 seeds would be enough to verify the assumptions required for a proper t-test? Particularly in experiments involving high variance models like neural networks. How and to what extent did you check the conditions for the t-tests associated with these reported p-values? And if the data is indeed suitable for t-testing, could you instead report confidence intervals in your plots instead of the standard deviation?  Then the conclusions reported in the text might better align what what can be seen at a glance in the plots, which currently give an impression that there is not a significant difference due to the overlapping measures of variance.
> >
> > In any case, taking the p-values at face-value for the moment, it's a bit unclear to me how significant the difference between AdaOPT and PLOT really is; while it might be true that there is a statistical difference, I get the impression from the plots that whatever potential difference might exist is negligibly small. However, it's hard to tell what constitutes a "significant" amount of additional regret here. It might be useful to include some more interpretable performance measure, to make it easier to assess the significance of improvement (e.g. a 0.1% improvement in classification accuracy on mnist can simultaneously be statistically significant and uninteresting)

---

> > > ### Author Response · Authors · 2022-11-19
> > > **Response to further comments by reviewer cz4a**
> > >
> > > Thanks for your answer.
> > >
> > > *I am not particularly familiar with all of these datasets, but I am a bit skeptical that 5 seeds would be enough to verify the assumptions required for a proper t-test?*
> > >
> > > As we wrote above and in the paper it is 5 runs over each of 5 seeds so 25 experiments altogether. Crucially these experiments are independent. We use t-test to evaluate the statistical significance of differences between the **final regrets** of each method. Central limit theorem implies that these means are normally distributed given large enough number of samples. Number of samples between 20 and 30 is universally considered enough for this assumption in the literature. Note that the above conditions are independent of the dataset (the properties of the datasets enter the calculations in t-test via their standard deviations; also for the purposes of the t-test in our case the datapoints of the datasets are the final regrets of each method, just in case this was a source of your confusion). Therefore the conditions for t-test are satisfied in our case.
> > >
> > >
> > >
> > > *And if the data is indeed suitable for t-testing, could you instead report confidence intervals in your plots instead of the standard deviation*
> > >
> > > Standard deviation carries important information about each individual method, e.g. the adversarial method on its own has a wide standard deviation. Your alternative suggestion has no clear interpretation, since confidence intervals can only be computed pairwise for the purposes of the t-test in the paper - so did you mean to suggest we plot e.g. 95% confidence intervals of each method compared to AdOpt? We think this would be confusing on the plot which has 6 different methods together. Instead, note that we make it clear in the experiments section that given the number of experiments we conduct (n=25) t-values greater than 2 in our experiments indicate less than 5% chance of error, which implies that the means are within non-intersecting 95% confidence intervals. (Or as in the case of Adult and MNIST 99.95%). We understand the point that the plots appeared confusing to you on the first glance (although if you view them on standard computer screen the means are very clearly separated), but the t-values are reported in the **Main results** section of the paper and it is a reasonable assumption that the reader of an ML paper possesses a level of statistical competence required to either immediately interpret these values or to do so after consulting a basic textbook.
> > >
> > > *I get the impression from the plots that whatever potential difference might exist is negligibly small. However, it's hard to tell what constitutes a "significant" amount of additional regret here.*
> > >
> > > Regret is the standard benchmark in the field (Zhou et al, Pacchiano et al etc) and the differences between mean final regrets of AdOpt and PLOT are in fact greater than between PLOT and NeuralUCB on Adult, whereas PLOT loses to NeuralUCB on MNIST (regrets are on Adult 37.23/39.78/40.73 on MNIST 3.746/4.529/4.095 for AdOpt/PLOT/NeuralUCB). The datasets we used in the paper are again standard benchmark datasets, on which the differences in regret on this scale are accepted in the literature to strongly indicate the superiority of the methods in question, in particular for wider class of real-world datasets. In practice even small difference in regret can have far-reaching implications (less unfair loan rejections, wrong treatments assigned to patients or even significant profits when trading stocks).

---

> > > > ### Comment · Reviewer_cz4a · 2022-11-19
> > > > **Response**
> > > >
> > > > >  Central limit theorem implies that these means are normally distributed given large enough number of samples. Number of samples between 20 and 30 is universally considered enough for this assumption in the literature
> > > >
> > > > How large the sample is really depends on underlying data distribution, and the 20-30 samples is only a rough rule of thumb, far be it from some universally accepted minimum sample size --- it is not a substitute for actually checking that the assumptions are met. Think about what's being reported: that *with high confidence* there is a difference in the measurements. Any such claim is only valid if the t-test itself was valid, which it frequently is not in real problems. One must use a different hypothesis test when this happens, t-testing is far from a catch-all
> > > >
> > > > > Note that the above conditions are independent of the dataset (the properties of the datasets enter the calculations in t-test via their standard deviations; also for the purposes of the t-test in our case the datapoints of the datasets are the final regrets of each method, just in case this was a source of your confusion)
> > > >
> > > > I'm not sure I follow, the assumptions required for t-testing depend on the underlying distribution, which effects how many samples would be needed to justify approximate normality.
> > > >
> > > > > Your alternative suggestion has no clear interpretation, since confidence intervals can only be computed pairwise for the purposes of the t-test in the paper
> > > >
> > > > Perhaps I phrased it confusingly; I'm not suggesting that you replace the t-tests with pair-wise confidence intervals. Rather, what I'm getting at is that if your data is nice enough to be performing t-tests on it, then it it likely also nice enough to compute confidence intervals around the mean as a measure of spread in your plots, rather than using the standard deviation. This would align with the t-tests in the sense that non-overlapping CI's would indicate a statistically significant separation, which is also what is reported in the main results.
> > > >
> > > > >Regret is the standard benchmark in the field (Zhou et al, Pacchiano et al etc)
> > > >
> > > > This does not really address the concern though; here the problem is that the difference between the proposed method and the primary baseline are fairly negligible in these experiments, and it's not clear how to assess the significance of this small gap in this case. Both Zhou et al. and Pacchiano et al. appear to suffer from the same issue, though I don't think that this excuses it as being an issue

---

> > > > > ### Author Response · Authors · 2022-11-19
> > > > > **Response to cz4a**
> > > > >
> > > > > You wrote *"How large the sample is really depends on underlying data distribution, and the 20-30 samples is only a rough rule of thumb, far be it from some universally accepted minimum sample size --- it is not a substitute for actually checking that the assumptions are met."*
> > > > > and also:
> > > > > *"I'm not sure I follow, the assumptions required for t-testing depend on the underlying distribution, which effects how many samples would be needed to justify approximate normality."*
> > > > >
> > > > > It was not clear what the issue is from your previous comment, so just to make sure we are talking about the same thing: t-test in our case is performed for the means. The distribution of the means converges to normal distribution as the number of samples grows. The rate of the convergence is estimated by a number of results in probability theory. In practice, t-test is used when the means are computed over 20-30 samples, since for this number of samples the distribution of the means is usually sufficiently close to normal for t-test to be meaningful. See e.g. Sawilowsky and Blair, 1992 that discusses the applicability of t-test for practical purposes across several distributions. However we are happy to conduct additional tests to verify that the means are approximately normally distributed, eg by computing the third and fourth moment, as well as increase the number of samples if you feel that in this particular case there is a concern that the convergence is unusually slow.
> > > > >
> > > > > **Although we can no longer edit the revision draft we are happy to share these results here. Do you have a concrete suggestion or preference for a test that would increase your confidence in the applicability of t-test in our case?**
> > > > >
> > > > > *Perhaps I phrased it confusingly; I'm not suggesting that you replace the t-tests with pair-wise confidence intervals. Rather, what I'm getting at is that if your data is nice enough to be performing t-tests on it, then it it likely also nice enough to compute confidence intervals around the mean as a measure of spread in your plots, rather than using the standard deviation.*
> > > > >
> > > > > Yes, if this is what you meant, we agree that it might provide a better representation of the data, and happy to change the plots accordingly in the next version of the paper.
> > > > >
> > > > > *This does not really address the concern though; here the problem is that the difference between the proposed method and the primary baseline are fairly negligible in these experiments, and it's not clear how to assess the significance of this small gap in this case. Both Zhou et al. and Pacchiano et al. appear to suffer from the same issue, though I don't think that this excuses it as being an issue*
> > > > >
> > > > > This is an accepted benchmark in the entire subfield. Perhaps a separate research work is needed to evaluate its meaningfulness but this is beyond the scope of the current work. I should point out again that (unlike improvements on MNIST) the practical problems these techniques aim to solve are of the kind where every improvement is meaningful in the *practical* sense, e.g. to provide fairer loans or identify life-saving treatments. This is an additional and separate dimension of "meaningfulness".
> > > > > **However if you would like to elaborate on your understanding of what meaningful improvement would mean in this case we would be very happy to discuss that.**

---

> ### Author Response · Authors · 2022-11-14
> **To the reviewer cz4a**
>
> Thanks again for reviewing our paper. We hope you had time to read our comments and the revised draft- could you please let us know if you have any additional questions or comments, especially regarding the empirical contribution, so we could address them before the the rebuttal period is over?

---

> ### Author Response · Authors · 2022-11-27
> **Re: your assessment of the empirical contributions**
>
> Since you have made no further clarifications to your queries in our discussion we would like to summarise our response to the points you made. We would be grateful if you could consider it:
> 1. You wrote *"...if your data is nice enough to be performing t-tests on it, then it it likely also nice enough to compute confidence intervals around the mean as a measure of spread in your plots, rather than using the standard deviation.*
>
> Here are the [plots](https://imgur.com/a/OUoMSdU) comparing AdOpt and PLOT as you suggested: with the 95% confidence interval around the means computed as $\pm SE*2.064$, using the t-value for 24 degrees of freedom. **Since this discussion started from the contradictory (in your eyes) appearance of the plots and stated t-values, if the new plots clear this confusion we would be grateful if you could revise your final score.**
>
> 2. You questioned that conditions for t-test are met *"How large the sample is really depends on underlying data distribution, and the 20-30 samples is only a rough rule of thumb, far be it from some universally accepted minimum sample size --- it is not a substitute for actually checking that the assumptions are met."*
>
> Just to clarify again that we are talking about the distribution of the means which by CLT is converging to normal distribution, and so t-test is applicable given large enough number of samples.
> The situation is very far from the *"rough rule of thumb"*: there is a line of work discussing applicability of t-test in the setting of non-normal distributions and relative to sample sizes. We already mentioned *Sawilowsky and Blair*; a more recent paper is e.g. *Lumley et al* and both contain numerous other citations. Here is a summary of relevant points:
> * Both papers mention the above sample size as being large enough for the t-test to be useful. More precisely for these sample sizes type I error rate is shown to be "conservative" for a wide variety of "abnormal" distributions considered (colloquially this means that falsely rejecting null hyphotesis is less or as likely than for normal distribution). This is just how we are using the t-test in our experiments.
> * Both papers mention skew or 3rd moment of the distribution as being the main issue impacting how close to normal is the distribution of the means: the higher the skew the more samples are needed for t-test to be close to what one would see if the distirbution was normal to begin with.
>
> Therefore we calculated the skew values for PLOT and AdOpt on all datasets. They are within the limits considered in the papers above.
> | Dataset | PLOT Skew |  AdOpt Skew |
> | -------- | -------- | -------- |
> | Adult     | 0.1     | 0.63     |
> | Bank     | 0.37     | 0.2      |
> | Crime     | 1.8     | 1.5      |
> | German     | 0.2     | 0.04     |
> | MNIST     | 0.3     | 1.4     |
>
> * Both papers find the distribution of the means to be close to normal at approximately this number of samples even for significantly higher skew values.
>
> Let us know if there was something else you wanted to clarify in this respect. It was possible to run more experiments to confirm the results but in the view of the above results from the literature and reported t-values it appears to be unlikely that there is any material doubt in the significance of our empirical findings.
>
> 3. We would like to reiterate our answer to what you write here: *"I get the impression from the plots that whatever potential difference might exist is negligibly small. However, it’s hard to tell what constitutes a “significant” amount of additional regret here.""* and also *Both Zhou et al. and Pacchiano et al. appear to suffer from the same issue, though I don't think that this excuses it as being an issue*
>
> The improvements in our paper are comparable with those reported by other published papers in the field, as we elaborate in our previous comments to you. You appear to question the entire line of works on the topic in your last sentence, contradicting the entire community of experts in this field - not sure whether this was your intention. A paper naturally should be judged against other peer-reviewed published research on the same topic - it would not be fair to review a paper in the field where you are in fact questioning or else aren't confident about the validity of accepted benchmarks.

---

> > ### Comment · Reviewer_cz4a · 2022-11-28
> > **Empirical contributions**
> >
> > I feel that important context is being ignored, so I will re-iterate: my issues with the empirical work have to do with several *related* things:
> > 1. The proposed method is a modification of an existing method, so it's important to show that there is actually some significant difference
> > 2. there is no rigorous theory showing that AdaOpt is *necessarily better* than PLOT, so the empirical evaluation is the only thing we can use to to assess the difference between these two methods
> > 3. The empirical work shows very little difference between the two, calling into question how significant the difference between the two really is.
> >
> > It is the *combination* of these three things that I find unconvincing. I don't think it unreasonable to expect a study which relies very heavily on its empirical evaluation to provide additional experiments or plot alternative measures of performance in the event of inconsistent or underwhelming evidence of an improvement in performance.
> >
> > >Here are the plots comparing AdOpt and PLOT as you suggested: with the 95% confidence interval around the means computed as , using the t-value for 24 degrees of freedom. Since this discussion started from the contradictory (in your eyes) appearance of the plots and stated t-values, if the new plots clear this confusion we would be grateful if you could revise your final score.
> >
> > These plots actually reinforce what I was saying earlier --- the confidence intervals are indeed overlapping 3/5 of the plots, suggesting no statistically significant difference in over half the experiments. Moreover, how is it that the t-test concludes a significant difference while 95% confidence intervals do not?
> >
> > >  there is a line of work discussing applicability of t-test in the setting of non-normal distributions and relative to sample sizes. We already mentioned Sawilowsky and Blair; a more recent paper is e.g. Lumley et al and both contain numerous other citations.
> >
> > I don't see why we should resort to relying on empirical observations for when it's "okay" to report invalid t-tests, especially given how extreme the performance distributions can be in deep learning problems in general. I'm not sure why it's necessary to speculate at all in this case --- the data is readily available, is it not? Why not just plot the empirical distribution to show that it is indeed approximately normal?
> >
> > > You appear to question the entire line of works on the topic in your last sentence, contradicting the entire community of experts in this field - not sure whether this was your intention
> >
> > Not sure what gives you this impression, criticizing a benchmark that a paper uses in their experiments doesn't dismiss the entire work. The issue in *this specific case* is that the assessment of novelty and significance of the proposed method are also tightly tied to the experiments. I also do not think it unreasonable to have a higher bar for novelty of the experimental results in general than prior works, given that the proposed method is a (novel) combination of *existing methods*.
> >
> > I remain unconvinced and maintain my score.

---

> > > ### Author Response · Authors · 2022-11-29
> > > **re: Empirical contributions part 1**
> > >
> > > Thank you for investing effort into engaging in the discussion - it had compelled us to find a better way to showcase our results. There is no disagreement between us that empirical validation has an important place in our paper, this was the starting point of our discussion. However we think that the analysis that lead you to to conclude that *"The empirical work shows very little difference* (between AdOpt and PLOT)" suffered from several innacuracies, in particular in your latest response. We detail the evidence supporting this statement below, and we also include results of additional experiments confirming that mean regrets have normal distributions, as you requested. We will be grateful if you could carefully examine and address these points accordingly.
> > >
> > > 1. **Our experiments show clear and statistically significant improvement on *Adult* and MNIST**. The improvement of AdOpt on these datasets is more than the comparable improvement of PLOT over the previous baseline in Pacchiano et al. In the [plots](https://imgur.com/a/OUoMSdU) we posted earlier following your request there is no intersection of 95%CI's.
> > > 2. Despite what you conclude in your last comment same [plots](https://imgur.com/a/OUoMSdU) also show that **there is clear and statistically significant improvement on *Crime* and a trend towards statistically significant improvement on *German***.
> > > Namely, you wrote:*"These plots actually reinforce what I was saying earlier --- the confidence intervals are indeed overlapping 3/5 of the plots, suggesting no statistically significant difference in over half the experiments. Moreover, how is it that the t-test concludes a significant difference while 95% confidence intervals do not?"*
> > > The fact that 95% CI's intersect for *Crime* and *German* doesn't imply that there is *"no statistically significant difference"* and doesn't contradict the t-test results. All experiments are independent hence probability that *both* means lie in the intersection of CI's is a product of probabilities for each one of the means to lie there. This probability is $0.0069=0.1*0.069$ for *Crime* and $0.14=0.45*0.31$ for *German*, which (unsurprisingly) aligns with what was predicted by t-test.
> > > To further illustrate this idea graphically we have made [mean difference plots](https://imgur.com/a/t7j7ZGW) of shaded 85% CI and 95%CI around the *difference* between the mean regrets of PLOT and AdOpt. These CI's correspond to the t-values reported in the paper. You can see that 95%CI doesn't include 0 on *Crime* and 85% CI doesn't include 0 on *German*, showing that the advantage of AdOpt is statistically significant. (We would like to remark that using our new experimental results with 10 seeds on which we elaborate in the next point we find that on *German* the level of statistical significance is actually 90%CI. In any case it shows a clear trend of improvement with increasing the length of adversarial training as we elaborate in Figure 4 of the revised draft.)
> > > 3. The only plot which shows no statistically significant difference is (as expected) plot for the dataset *Bank*. It is ok - we do not cherry-pick our empirical results.
> > >
> > > We hope the above clarifies any doubt you might have about inconsistences between plots and reported t-values and confirms that t-test indeed indicates significant difference between the mean regrets on 4 out of 5 datasets considered.
> > >
> > > Since you still question the applicability of t-test to our data, let us address this next. You wrote: *I don't see why we should resort to relying on empirical observations for when it's "okay" to report invalid t-tests, especially given how extreme the performance distributions can be in deep learning problems in general. I'm not sure why it's necessary to speculate at all in this case --- the data is readily available, is it not? Why not just plot the empirical distribution to show that it is indeed approximately normal?*
> > > Here is the [histogram plot](https://imgur.com/a/60ubzWL) of the distribution of the means over 25 experiments on *Adult* for all methods considered. It was computed using Monte Carlo method and the results of experiments with the additional seeds (50 experiments in total) we have run specifically for this objective in light of this discussion. You can see for yourself that these are Gaussian distributions.
> > >
> > > *response continues in the next comment*

---

> > > > ### Comment · Reviewer_cz4a · 2022-11-30
> > > > **Re:Re:empirical contributions parts1/2**
> > > >
> > > > > Our experiments show clear and statistically significant improvement on Adult and MNISTOur experiments show clear and statistically significant improvement on Adult and MNIST
> > > >
> > > > I agree, my issue was specifically with the other 3
> > > >
> > > > > Here is the histogram plot of the distribution of the means over 25 experiments on Adult for all methods considered
> > > >
> > > > Thank you, I find this significantly more convincing than the previous arguments
> > > >
> > > > >The fact that 95% CI's intersect for Crime and German doesn't imply that there is "no statistically significant difference" and doesn't contradict the t-test results
> > > >
> > > > You are correct, this was careless of me; my apologies. Overlapping CI's do not necessarily imply failure to reject the null hypothesis.
> > > >
> > > > The result still indicates insignificant difference on two of the three I had mentioned, and a potentially small but significant difference
> > > > on crime. I generally do not agree with changing the significance level post-hoc to support your conclusions in the German dataset. So overall the conclusions are more or less the same as what one would conclude from overlapping CI's, with either insignificant or potentially weak results in 3 out of 5 experiments.
> > > >
> > > > This goes back to what I had said earlier, that there isn't any good way to gauge significance of the differences here; even on mnist where there's a consistent statistically significant difference, the difference in regret is less than 1; is that large? small? I don't know, and there's no alternative metrics showing a more clear difference in performance. Why should I assume that a difference in regret of 1 is large, rather than assume it's small? You argue that even small differences can lead to large impacts: earlier it was argued that
> > > >
> > > > > even small difference in regret can have far-reaching implications (less unfair loan rejections, wrong treatments assigned to patients or even significant profits when trading stocks)
> > > >
> > > > But I don't find this convincing at all. One can apply this argument to nearly any algorithm, by claiming they plan to use it for something important. It doesn't change the fact that there's no real way to judge the measure of scale here, which is particularly relevant when the difference *appears to be* rather small, and there's no attempt to dig deeper into the results to show otherwise, be it through more informative additional metrics or additional experiments.
> > > >
> > > > >Both papers we mentioned strongly rely on the fact that their proposed approach empirically improves the same benchmark that we use, cumulative regret
> > > >
> > > > Let me clarify, as I can see how my comment could come off the wrong way. I was pointing out that my criticism of your and previous works benchmarks does not dismiss the contributions of prior works. Then, I was arguing that in *this specific case* the empirical work was not convincing enough (in my opinion) to provide novel empirical insights, and moreover, the unconvincing experiments make it hard to assess the significance of the proposed changes. This does not imply that I think that the prior works necessarily did or did not have more novel algorithms or contributions one way or the other. My point is merely that I think that, given that this method is a combination of existing methods and has no rigorous theory or guarantees, there is a heavier burden placed on the empirical work to provide novel insights and demonstrate significant advantage over the baseline, which I do not believe that the current version of this work provides. In my opinion, the results presented in the current work are not strong enough to provide enough insight to put it above the acceptance threshold --- what I think about the prior works is irrelevant here and doesn't change the fact that I find the results presented unconvincing.

---

> > > > > ### Author Response · Authors · 2022-12-02
> > > > > **re: empirical contribution**
> > > > >
> > > > > Thanks for the response. We understand that following our discussion you now agree that t-test is applicable in the setting of our experiments and indicates that our results are statistically significant. (Here are mean distribution [plots](https://imgur.com/a/uzlLr2t) for the remaining datasets to justify t-test applicability for them as well). This is a very important point since we regard the robust validation of statistical significance as the most important novel *empirical* contribution of our work.
> > > > >
> > > > > You write *"My point is merely that I think that, given that this method is a combination of existing methods and has no rigorous theory or guarantees, there is a heavier burden placed on the empirical work to provide novel insights and demonstrate significant advantage over the baseline, which I do not believe that the current version of this work provides."*,
> > > > >
> > > > > You seem to ignore our entire discussion in this assessment - in contrast to previous works our paper reports results over several seeds and assesses their statistical significance using a method we both now agree is robust.
> > > > >
> > > > > This rigorous evaluation extends to the previous benchmark results: For example, our experiments show that when carefully evaluated across several seeds PLOT (combined with epsilon-greedy as in Pacchiano et al) doesn't in fact outperform the previous benchmark, NeuralUCB on neither MNIST nor Bank and it's improvement on Adult is very small compared to AdOpt. In contrast to that AdOpt outperforms by a **large margin** on 2 out of 3 among these datasets. The conclusion is that the choice of pseudo-label candidates in AdOpt significantly improves performance on the datasets tested by Pacchiano et al.
> > > > >
> > > > > It seems that the only question that you still have about our results is indeed how to interpret the improvement in performance. You write: *This goes back to what I had said earlier, that there isn't any good way to gauge significance of the differences here; even on mnist where there's a consistent statistically significant difference, the difference in regret is less than 1; is that large? small? I don't know, and there's no alternative metrics showing a more clear difference in performance.*
> > > > > There is in fact a simple way to measure the improvement quantitatively: to introduce a scale against which this improvement should be evaluated one can take the performance of the *Greedy* method as a baseline, and then calculate amount, expressed as percentage by which different algorithms improve over this baseline. Here are the results.
> > > > > | Dataset | Greedy Mean |PLOT improvement |AdOpt Improvement|
> > > > > | -------- | -------- | -------- |-------
> > > > > | Adult     | 40.43     |2%     |9%
> > > > > | MNIST     | 4.46  |4%      |18%|
> > > > > | Crime     | 23.97     | 48%      |54%   |
> > > > > | German     | 32.86    | 43%     |46%
> > > > > | Bank     | 25.79   |5%  | 7%     |   6%|
> > > > >
> > > > > Reducing regret with respect to to Greedy method is a basic requirement of any algorithm in the field. We see that on Adult and MNIST AdOpt reduces regret from baseline 4.5 times more than PLOT, and the while improvement over Crime and German isn't as dramatic AdOpt still experiences 6% and 3% less regret than PLOT on average.
> > > > >
> > > > > This should clear any doubt about the magnitude of our emprical improvement on Adult and MNIST. Re: Crime and German, we maintain that several percentages improvement in regret that **importantly** is shown to be statistically significant are a meaningful improvement of potentially big practical significance (e.g. *average* 3 percent less wrong rejections over many loans given is a big number) that merit publication.
> > > > >
> > > > > We hope we managed to demostrate that our results are different from your analogy with 1% improvement in digit recognition on MNIST, which is **small relative to benchmark improvement** and is likely to be insignificant due to noise in either dataset itself or the model and whose practical significance is also questionable. None of these points are true in our case.
> > > > >
> > > > > We appreciate the discussion we have had up to now, that had lead us to add add to and improve the presentation of the empirical results clearing possible misconceptions. Your initial assessment was based in questioning the validity of t-test and stating *"there is no statistically significant difference between PLOT and AdaOpt on any of the datasets except for "Adult"* However given the explanations we gave above we do not understand how you can still maintain that empirical contribution of our paper can be assessed as *"Neither significant nor novel"*. It conducts order of magnitude more experiments, it re-evaluates previous results and it achieves 3 to 5 times more improvement in performance over previous method on some datasets, while also meaningfully improving on two others (that weren't even considered in previous work)!
> > > > >
> > > > > We would be grateful for your further comments.

---

> > > > > > ### Comment · Reviewer_cz4a · 2022-12-03
> > > > > > **re:empirical contribution**
> > > > > >
> > > > > > > We understand that following our discussion you now agree that t-test is applicable in the setting of our experiments and indicates that our results are statistically significant.
> > > > > >
> > > > > > This is not an accurate summary of what I said. I said I agree that the t-test is applicable based on the histograms, but that my conclusions about the experiments were effectively unchanged from before.
> > > > > >
> > > > > >  Again, I do not consider the result for German significant --- your test clearly shows there was no significant difference so an additional test was computed at a lower confidence so that it could conform with the pre-conceived conclusion. By this same methodology, I could go back and reject all the experiments by post-hoc changing to a higher confidence level.
> > > > > >
> > > > > > Hence, I do not consider the results for Bank nor German to be significant, and the third result, Crime, shows a significant but unsubstantial difference, which is further reinforced by the table provided (see also below). Overall more than half of the results fail to demonstrate a clear or meaningful improvement over the baseline in my opinion.
> > > > > >
> > > > > > > You seem to ignore our entire discussion in this assessment - in contrast to previous works our paper reports results over several seeds and assesses their statistical significance using a method we both now agree is robust.
> > > > > >
> > > > > > I respectfully disagree. Even if we take the statistical significance at face-value and ignore measures of spread altogether, I do not believe that the results presented demonstrate a clear or meaningful improvement over the key baseline, for the reasons I've already stated.
> > > > > >
> > > > > >
> > > > > >
> > > > > > > we maintain that several percentages improvement in regret that importantly is shown to be statistically significant are a meaningful improvement of potentially big practical significance (e.g. average 3 percent less wrong rejections over many loans given is a big number) that merit publication
> > > > > >
> > > > > > I maintain that this is not a compelling argument, it's always possible to argue that "small differences matter because our algorithm could be used in a high-impact setting", and this is not unique to the bank-loan problem. But even if we take this argument as given, the fact of the matter is that this *isn't* such a high-impact situation --- these are ultimately small/"toy" examples and the whole point of them is to demonstrate the claims made in the paper and provide insight, which they are not doing very clearly. So while a few percentages may matter in the applications you mention, they do not matter so much here, where the experiments are *specifically designed to support the claims made in the paper*.
> > > > > >
> > > > > > >There is in fact a simple way to measure the improvement quantitatively: to introduce a scale against which this improvement should be evaluated one can take the performance of the Greedy method as a baseline
> > > > > >
> > > > > > Yes, something of this nature would be more convincing to me and could potentially significantly improve the paper, though I still find this  particular table unconvincing because measures of spread again become a concern. The differences are marginal in all experiments except for the ones that we already agree show substantial improvement (Adult and MNIST), and on top of that we again can't claim that these small differences are actually statistically significant without further investigation. In general even for independent random variables $\mathbb{E}[X/Y]\ne \mathbb{E}[X]/\mathbb{E}[Y]$, so even if we were to *assume* the previous results were statistically significant it does not imply that the results in this table are.
> > > > > >
> > > > > > >Your initial assessment was based in questioning the validity of t-test and stating "there is no statistically significant difference between PLOT and AdaOpt on any of the datasets except for "Adult" However given the explanations we gave above we do not understand how you can still maintain that empirical contribution of our paper can be assessed as "Neither significant nor novel". It conducts order of magnitude more experiments, it re-evaluates previous results and it achieves 3 to 5 times more improvement in performance over previous method on some datasets, while also meaningfully improving on two others (that weren't even considered in previous work)!
> > > > > >
> > > > > > This is another mischaracterization of what I've said. Validity of the t-test was a concern, yes, but  was only *one* of my issues with the experiments, and I do not agree with the items in the subsequent list were done convincingly enough to constitute a novel or significant empirical result for reasons I've already stated.
> > > > > >
> > > > > > Overall I think the new table provided is a promising direction that could improve the quality of a future submission. My main suggestions for a future submission would be to focus the evaluation more on *clearly* demonstrating differences from the baseline, especially in situations where the existing benchmark provides insufficient evidence.

---

> > > > > > > ### Author Response · Authors · 2022-12-03
> > > > > > > **re:empirical contribution**
> > > > > > >
> > > > > > > Thanks for the reply. Although it doesn't seem likely that it would change your assesment let us address a few points in your comment.
> > > > > > >
> > > > > > > >This is not an accurate summary of what I said. I said I agree that the t-test is applicable based on the histograms, but that my conclusions about the experiments were effectively unchanged from before.
> > > > > > >
> > > > > > > We don't follow you here - if t-test is valid and indicates statistical significance of results then results are statistically significant. This is what this sentence says. Your conclusions about the experiments are clearly rooted in something other than their statistical significance at this point in the discussion.
> > > > > > >
> > > > > > > >Again, I do not consider the result for German significant --- your test clearly shows there was no significant difference so an additional test was computed at a lower confidence so that it could conform with the pre-conceived conclusion.
> > > > > > >
> > > > > > > We did not perform any additional testing. We simply displayed graphically the level of cofidence the test implies, which in this case was about 85% for 25 seeds and 90% for 50 (which we run to provide you with the histograms for the means distribution). The 95% confidence interval is obviously displayed for comparison with other methods. It is up to you whether you consider 85% or 90% confidence interval meaningful, the information is simply there for your or anyone else's interpretation.
> > > > > > >
> > > > > > > >Yes, something of this nature would be more convincing to me and could potentially significantly improve the paper, though I still find this particular table unconvincing because measures of spread again become a concern. The differences are marginal in all experiments except for the ones that we already agree show substantial improvement (Adult and MNIST), and on top of that we again can't claim that these small differences are actually statistically significant without further investigation. In general even for independent random variables $E(X/Y)\neq E(X)/E(Y)$, so even if we were to assume the previous results were statistically significant it does not imply that the results in this table are.
> > > > > > >
> > > > > > > We are glad you found the table useful.
> > > > > > >
> > > > > > > Estimates of expectation and variance of the ratio of independent normal variables are well studied in the literature and can be easily computed in this case to show statistical significance of the ratios of improvement.
> > > > > > >
> > > > > > > What do you mean again by *so even if we were to assume the previous results were statistically significant?* Those results *are* statistically significant according to t-test.
> > > > > > >
> > > > > > > Our impression at this point is that you question the magnitude of improvement on German, Crime and Bank. However this doesn't justify your assessment of our empirical results as "neither novel nor significant" - since by now as a result of our discussion you have agreed that in contrast to previous works our paper reports results over several seeds and assesses their statistical significance using t-test. This is by definition novel, and since we agree on improvement on MNIST and Adult also (at least) partially significant.

---

> > > ### Author Response · Authors · 2022-11-29
> > > **re:Empirical contributions part 2**
> > >
> > > Why did we have to run new experiments to establish the distribution of the means? Let us repeat again: we are interested in the distribution of the **mean regrets**, not the **individual regrets for each run**. The distribution of individual regrets may well not be normal but by CLT the distribution of mean regrets will converge to normal. 25 experiments that we had run initially do not allow us to reliably estimate the *distribution* of the means. So when you said *"...the data is readily available, is it not?""* - the answer is *"No, unless you run more experiments, it isn't"*. For that reason in our previous responses we relied on the literature that strongly suggests that 25 experiments would make the distribution of the means sufficiently close to normal. We also did check that the distribution of individual regrets isn't *"wild"* as you suggest, by estimating their skew, which is what mainly influences t-test errors. By the way just this precise situation is the subject of the papers we cited in previous comments.
> > >
> > > We would like to address your last point: *"Not sure what gives you this impression, criticising a benchmark that a paper uses in their experiments doesn't dismiss the entire work. The issue in this specific case is that the assessment of novelty and significance of the proposed method are also tightly tied to the experiments. I also do not think it unreasonable to have a higher bar for novelty of the experimental results in general than prior works, given that the proposed method is a (novel) combination of existing methods."*
> > > Both papers we mentioned strongly rely on the fact that their proposed approach empirically improves the same benchmark that we use, cumulative regret. If you still see a reason that our paper should be subjected to different criteria in terms of empirical results, you need to specify them concretely. Note however that we did in fact run order of magnitude more experiments in order to establish statistical significance of our results and added two datasets compared to Pacchiano et al, which you consider our main benchmark.
> > >
> > > We hope that the improved presentation in the plots as well as our explanation above have cleared whatever misunderstanding there was regarding the statistical significance and the use of t-test and happy to answer any additional questions.

---

> ### Author Response · Authors · 2022-12-03
> **to reviewer cz4a**
>
> We feel that we addressed the points you have raised to the best of our ability - thanks again for actively engaging in the discussion.

---

### Official Review · Reviewer_4jYB · 2022-10-24

**Confidence:** 2
**Correctness:** 3
**Technical Novelty And Significance:** 3
**Empirical Novelty And Significance:** 3
**Recommendation:** 6

**Clarity, Quality, Novelty And Reproducibility:**

Clarity and Quality: The writing is excellent and easy to follow. The literature review is cut short compared to PLOT. I personally believe that this paper can attract a larger audience by including the discussion on its parallel to strategic classification and performative prediction.

Novelty: Half of the paper (adversarial domain adaptation) is new contribution compared to PLOT.

Reproducibility: I briefly read the code. It appears to be well-written and documented.

**Strength And Weaknesses:**

=== Strength ===

According to the authors and the best of my knowledge, this paper is the first attempt to utilize adversarial domain adaptation to tackle bias in online learning. The paper is well-written and easy to follow.

=== Weaknesses ===

The paper refers to G as a de-biased generator and C a de-biased classifier, but there is no discussion in 3.3 on what "de-biased" means, or how exactly adversarial domain adaptation achieves it.

If the role of the de-biased classifier, as expected, turns out to be trading off precision and recall (compared to the biased classifier), why don't we just do thresholding on the biased classifier? That is, go with PLOT, but simply choose a good threshold (<.5) in the biased classifier to select nodes to assign pseudo-labels. In any case, a precision-recall-curve comparison is necessary to establish that the de-biased classifier is trading off precision and recall at a larger area-under-the-precision-recall-curve (compared to the biased classifier) in the threshold region of interest, but this is missing from the paper.

The paper also lacks an evaluation of the de-biased classifier: how unbiased is it actually, over the time steps?

**Summary Of The Paper:**

PLOT [Pacchiano 2021] leverages a adversarial-training-like scheme to train NN for online binary classification problem: at each time step, retrain the model M while pretending that the new data are of the positive class, and give prediction on the new data using the retrained model M.

This paper proposes to add a second classifier C to this training process. C is trained at each time step using adversarial domain adaptation; it classifies the embedding of the data. Now, only those that are rejected by M and accepted by C enters the PLOT training for this time step.



**Summary Of The Review:**

The paper is well-written, the contribution is non-trivial, and should be of interest to a broad online learning / strategic learning / domain adaptation audience. However, due to missing important ablation studies, I recommend reject.

---

> ### Author Response · Authors · 2022-11-10
> **Response to reviewer 4jYB**
>
> Thank you for reading the paper. We appreciate your assessment that the paper makes significant contribution to the existing field of knowledge. However we disagree with your assessment: *Half of the paper (adversarial domain adaptation) is new contribution compared to PLOT.*
>
> The use of adversarial domain adaptation in the online setting is our main contribution and in fact the main focus of our *whole* paper. We *use* the existing PLOT method as strategy for incorporating optimism into a DNN model using pseudo-labels. As we explain in the Introduction, for achieving SOTA performance in terms of regret PLOT has to be combined with an additional strategy for choosing pseudo-label candidates. Our method, AdOpt, offers a data driven aproach for the selection of pseudo-label candidates, by using adversarial domain adaptation. By combining domain adaptation with the pseudo-label approach of PLOT we are able to achieve statistically significant improvement in performance on several benchmark datasets.

---

> > ### Author Response · Authors · 2022-11-10
> > **Continuation of Response to 4jYB**
> >
> > We note that the majority of your questions stem from the misunderstanding of the use of the word "de-biased" in the context of the paper, and also possibly not being fully familiar with the existing research on domain adaptation. We hope to remedy this below:
> >
> > You wrote *The paper refers to G as a de-biased generator and C a de-biased classifier, but there is no discussion in 3.3 on what "de-biased" means, or how exactly adversarial domain adaptation achieves it*
> > The use of "biased" and "de-biased" terminology was introduced in Zhang et al. There it is used in the context of producing a representation of the data that makes it harder to distinguish between two chosen subsets of the data, e.g. between males and females. For the AdOpt method the subsets in question are the set of accepted applicants and the incoming batch. The de-biased generator is the NN that produces such train/test agnostic (de-biased) representation and the de-biased classifier is (simultaneously) trained on that representation to minimize the loss from Equation 4. A hallmark of de-biased classifier is demographic parity of predictions (Equation 5) (see e.g. Ganin et al, Edwards&Storkey) and the last three columns in Table 1 shows empirically that the classifier in AdOpt satisfies this equation. To reiterate the biased/de-biased terminology in the paper is used in the above precise sense and not "colloquially".
> >
> > We think that the second part of your question is just due to misunderstanding of the above terminology, but will add - just in case - that producing de-biased representation in the above sense is the whole objective of adversarial training and the entire subfield is devoted to the study of this subject (we cited a few of the most prominent papers in our work).
> >
> > We have included the explanation of the "biased/de-biased" terminology in the section 3.3 of new draft of the paper - we do think that it is an important clarification for the wider reader audience, so thank you very much for discussing it.
> >
> > **Complimentary to the above** we do like to note here that we have conducted some initial evaluations that indicate that the adversarial training produces a classifier, that is also "unbiased" in the more conventional sense, in that it treats the under-represented groups more fairly than other methods. The relevant data is in Figure 5 of our paper. The complete exploration of this very interesting phenomenon is a subject of entirely separate potential follow up work.

---

> > > ### Author Response · Authors · 2022-11-10
> > > **Continuation of Response to 4jYB**
> > >
> > > You wrote *If the role of the de-biased classifier, as expected, turns out to be trading off precision and recall (compared to the biased classifier), why don't we just do thresholding on the biased classifier? That is, go with PLOT, but simply choose a good threshold (<.5) in the biased classifier to select nodes to assign pseudo-labels. In any case, a precision-recall-curve comparison is necessary to establish that the de-biased classifier is trading off precision and recall at a larger area-under-the-precision-recall-curve (compared to the biased classifier) in the threshold region of interest, but this is missing from the paper.*
> > >
> > > Since our incoming batch is **unlabelled**, it is not clear how to chose such "a good threshold" in the biased classifier. E.g choosing the threshold to be zero would raise recall to 1 with the expected effect on precision, but (as shown in the article by Pacchiano et al) such a method of choosing pseudo-labels (i.e. simply taking the entire batch) doesn't produce competitive performance cumulatively in terms of regret. A "good threshold" is therefore one that combines "good" recall and precision. However since the *biased classifier* is trained on the population of accepted applicants, whose distribution is different from the distribution of the incoming batch, it is not clear why would one expect such a "good threshold" to exist at all. Optimising performance (or more specifically finding an optimal point on the precision/recall curve) in the presence of distributional shift is a fundamental question in machine learning, and is in fact precisely one of the main objectives of domain adaptation as a whole. There the optimal threshold is essentially learned with the help of the unlabelled data. This is exactly the reason for the use of domain adaptation in this paper. Specifically we use the adversarial approach here. This is done repeatedly for every incoming batch. We note again, that knowing AUC of the adversarially trained classifier doesn't help us to choose an appropriate threshold on unlabelled data.
> > >
> > > There are numerous papers comparing the adversarially trained classifiers with a baseline for classification purposes(in particular for the DANN method we use here) and these results would hold for every batch in our setting. Therefore we disagree that e.g. the precision-recall analysis that you suggest in our paper would add anything new to the existing knowledge in that respect.
> > > In contrast to that **the cumulative analysis** of the empirical performance of de-biased classifier is of significance for our online problem. The novelty of our method is in applying domain adaptation in the online setting. Accordingly we conduct this analysis in Table 1. We check that the recall is increased and precision is above what would result from picking examples randomly and crucially we show that it improves performance in terms of regret in Figures 2, 3, and 4.
> > >
> > > You wrote *The paper also lacks an evaluation of the de-biased classifier: how unbiased is it actually, over the time steps?*
> > >
> > > In terms of the meaning of the word *unbiased* as it is used in the paper (i.e. unbiased to being from test/train domain) we explained above that such an evaluation is the subject of the "predicted positives" section in Table 1. In the more conventional colloquial sense of the word, although there is no agreed definition of "unbiased", we show some promising initial results in Figure 5.
> > >
> > > *You mention the missing "ablation studies".* The two natural ablation studies for AdOpt are the evaluation agains PLOT and against standalone adversarial classifier, and both are included in our paper.
> > >
> > > Finally, we agree that the strategic classification and performative prediction are closely related areas and have included the discussion in literature review. Thanks for pointing this out.

---

> > > > ### Comment · Reviewer_4jYB · 2022-11-11
> > > > **Response to the authors**
> > > >
> > > > I sincerely apologize for "quantifying" the novelty of the paper. I thought by "half" I meant it as a compliment, as reflected in my novelty score: The contributions are significant and somewhat new. Aspects of the contributions exist in prior work.
> > > >
> > > > Thank you for clarifying the notion of de-biasing. It does help me better understand the paper. However, correct me if I am wrong, I think my last point still stands: it should be possible to evaluated the biased-ness of the de-biased generator, perhaps with the help of an oracle classifier. It may be the case that such study on de-biasing is not the focus of the paper and was been done in earlier works on de-biasing, but it does left me wondering. Nonetheless, I will lower my confidence score to a 2 in case an earlier work of this nature has been done and is properly cited in the paper.
> > > >
> > > > On precision-recall:
> > > >
> > > > The problem at hand is fundamentally an imbalanced learning problem, with the special property that the imbalance is iteratively introduced (and handled) by the classifier itself. Two classical approach to imbalanced learning are resampling and thresholding. Ideally, an imbalanced learning paper should compare to at least one of the two. A natural approach to this imbalanced learning problem is to deploy a decreasingly conservative classifier, and thresholding is perhaps the simplest way to achieve said conservatism. I believe the awareness of such goal is why the paper discusses precision-recall to begin with.
> > > >
> > > > When the de-biased classifier at threshold .5 appears to be trading precision and recall compared to the biased classifier at threshold .5, we want to be sure that it is not the case of this [imgur.com/a/KGbemVY] throughout timesteps. Here, the red point is the threshold =.5 point of the PR curve of the biased classifier; blue is that of the de-biased. If this was the case throughout timesteps, then 1) we would have been able to improve upon the biased classifier simply through thresholding and 2) it is possible that the optimal threshold of the biased classifier isn't the blue point either. Finding and comparing to this threshold (either as a constant or as a schedule) may be difficult, but an analysis of the PR curve is the first step to eliminate the necessity to make this comparison.
> > > >
> > > > It is possible, like the authors suggested, that the result of such comparison to thresholding is trivially known to readers familiar with the literature. For this reason, I lower my confidence score to 2, and ask the authors to kindly provide a reference to existing knowledge related to this comparison.

---

> > > > > ### Author Response · Authors · 2022-11-12
> > > > > **Response to reviewer 4jYB**
> > > > >
> > > > > Thanks for your through response, and clarifying again your assessment of the novelty of our paper. We were not sure how to interpret what you wrote since you gave the paper a score of 3, which we find contradictory to your assessment that it makes significant new contributions. Could you please clarify if the reason for your score was just what you consider to be the missing ablation studies?
> > > > >
> > > > > **Re: your comment on de-biasing**: the de-biased generator is a part of the adversarial triad of classifier, generator and discriminator trained to minimize the loss in equation 4. The discriminator and generator are working against each other: the former is trained to distinguish between the representations of source and target data created by the latter. The objective of adversarial de-biasing is to train a classifier that makes similar predictions for source and target data, i.e. exhibits demographic parity as in equation 5 in our paper. The work of Ganin et al that first introduced the method of adversarial domain adaptation conducts empirical evaluations of the "de-biasedness" of the generator and the classifier using several approaches, and in particular in terms of the generator creating indistinguishable representations of source and target using oracle discriminator as you suggested. However most of the literature uses the approach of evaluating whether "de-biasing" causes classifier and generator to satisfy "demographic parity" (equation 5 in our paper), since it also measures the performance of the de-biased classifier, whose performance is most pertinent here. For example, the work of Edwards and Storkey which uses a setup similar to the one in Ganin and in our paper evaluates this the de-biasedness in this sense on the Adult dataset with respect to several source/target splits. Therefore in our work, that deals with the *online* application of this method we also evaluated equation 5. We presented the cumulative statistic (means and std's) on proportion of predicted positives in the last two columns of Table 1, but our algorithm in fact generates these numbers for every batch, as we note in Section 5. Last two columns of Table 1 show that de-biased classifier predicts approximately same proportion of positives on the target (incoming batch) as it does on the source set $\mathcal{S}$ (accepted applicants). We hope it clarifies this issue!
> > > > >
> > > > > **Re: precision-recall**:
> > > > > You wrote: *The problem at hand fundamentally an imbalanced learning problem, with the special property that the imbalance is iteratively introduced (and handled) by the classifier itself.*
> > > > >
> > > > > We disagree with this assessment. The fundamental problem we are trying to solve is not the class imbalance problem, but a problem of predicting in the presence of covariate shift between source and target domains. The source of bias in the BLP are self-reinforcing false rejects. This means that candidates with certain features are underrepresented or altogether absent in the source domain. For example, the algorithm might stop accepting applicants living at a certain postcode after encountering several defaults from such applicants. To counteract this we make algorithm reconsider some of the rejected applicants at each step. This is a general approach to this issue in the wide body of literature on reinforcement learning, in particular bandits. Varying threshold cannot achieve this objective: since for example for certain features we only have samples labelled {'reject'} in the source domain. Simple thresholding is not an effective solution to this class of problems, so no direct comparisons is likely to exist in the literature. The closest method to thresholding in this setting would be perhaps distribution matching, eg Huang et al that was compared to eg DANN by Ganin et al.
> > > > >
> > > > > There is otherwise a multitude of domain adaptation methods proposed lately to approach this type of problems, and in this paper we are concentrating on a specific one, adversarial domain adaptation. We show empirically that it manages to reduce regret compared to other existing methods for introducing optimism and hence conclude that it is successful in solving the false rejects problem. We hope it is clear from the above discussion that thresholding does not constitute an ablation study for our method.

---

> > > > > > ### Comment · Reviewer_4jYB · 2022-11-14
> > > > > > **Response to the authors**
> > > > > >
> > > > > > On de-biasing: thank you for the clarification. Somehow I didn't connect demographic parity with "de-biasedness". Perhaps integrating some of your writing here into the paper can help readers not familiar with the literature. I have raised my score to a 5.
> > > > > >
> > > > > > On precision-recall and ablation study: the ablation study I have in mind is thresholding. In my opinion, the motivation to compare to thresholding arises when 1) the goal is to train a "conservative" classifier and 2) the de-biased classifier turns out to be trading precision for recall. Thresholding becomes a necessary ablation study when the observation about precision-recall is made, regardless if your proposed method is more SelectiveNet [Geifman 2019] than thresholding. Simple thresholding (or sophisticated ones, depending on how strong you want your baseline to be) may not be an effective solution to this class of problems, but you have to show it by ruling out this [imgur.com/a/KGbemVY] scenario.

---

> > > > > > > ### Author Response · Authors · 2022-11-17
> > > > > > > **Response to reviewer 4jYB**
> > > > > > >
> > > > > > > **Re: de-biasing and demographic parity:**
> > > > > > > Thanks for sincerely engaging in the discussion, we were happy to be able to explain these points. Due to space constraints in the current draft we added a short essential explanation of the "de-biased" terminology in section 3.3 as you suggested, also note that demographic parity property and it's empirical evaluation is already discussed in detail in 4.1. We agree that more detailed explanation can be beneficial and will include it in the camera-ready version.
> > > > > > >
> > > > > > > **Re:thresholding.** You wrote: *Thresholding becomes a necessary ablation study when the observation about precision-recall is made*
> > > > > > >
> > > > > > > We get your point. We have made [plots](https://imgur.com/a/hiIBlvc) (please click the link) comparing the regret of AdOpt with PLOT combined with biased classifier for pseudo-label selection at thresholds of 0.1 and 0.25 on the Adult, MNIST and Crime datasets. These plots show that AdOpt outperforms by a significant margin. These plots are computed over 5 different seeds as with all other experiments in our paper.
> > > > > > >
> > > > > > > We would like to remark, that to our knowledge thresholding was not previously tried as an approach to BLP or similar online problems. The reason for that in our opinion is that in the absence of labelled data from the target domain choosing the correct threshold is a non-trivial task, as we indicated before. Therefore while It is possible that exploring alternative thresholding regimes can be of interest (especially in light of what we show with AdOpt in our current work), this would constitute a novel and separate research project. In particular we would expect a "good" threshold to be dataset dependent, which then would require some calibration regime to work well in an online setting.
> > > > > > >
> > > > > > > We are happy to answer any additional questions you might have.

---

> ### Author Response · Authors · 2022-11-27
> **to reviewer 4jYB**
>
> Thanks for the productive discussion. Could you indicate whether you have looked at the thresholding ablation [plots](https://imgur.com/a/hiIBlvc)? Do they address the remaining point of your review, and if so, could you please consider raising your score?

---

> > ### Comment · Reviewer_4jYB · 2022-11-28
> > **Response to the author**
> >
> > Thank you for addressing my questions. I have raised my score to a 6. I think the plots you added is a good start to investigate the conservativeness of the learned classifier. A more thorough study would involve actually looking at the PR curve and the AUPRC at the high recall region.

---

### Official Review · Reviewer_ktkm · 2022-10-25

**Confidence:** 3
**Correctness:** 3
**Technical Novelty And Significance:** 2
**Empirical Novelty And Significance:** 3
**Recommendation:** 6

**Clarity, Quality, Novelty And Reproducibility:**

I find that the paper is easy to follow.

The submission includes python codes but due to time constraints, I could not verify whether the results are reproducible.


Minor comments:
-Section 1, Paragraph 8: mitigate - remove “for”
-Section 2, Paragraph 6: remove “in”
-Section 4, Paragraph 1: lables -> labels
-Section 4, Paragraph 3: approximating -> approximate
-Section 4.1, Paragraph 1: posses -> possess
-Section 4.2, Paragraph 2: picking -> pick
-Equation 3, Line 1: (\x_0, 1)
-Equation 4: not specify the set of x for the third term


**Strength And Weaknesses:**

Strengths:
-The paper concentrates on an interesting and significant problem characterized as the Bank Loan Problem.
-The paper is easy to follow.

Weaknesses:
1. The use of adversarial domain adaptation in the paper is not conceivable. The adversarial domain adaptation aims to equalize the probability of 1s from both domains (Equation 5). However, it inherently contradicts the PLOT method (Step 5, Algorithm 1).
2. It is unclear how the debiased classifier's performance impacts the learners' final performance compared to the previous method (PLOT). It would be valuable if the authors could provide a visualization of the proposed method (see Figure 2 in the PLOT paper).
3. The proposed approach requires training an additional debiased classifier in comparison to PLOT. However, I am not convinced that the experimental results in Figure 2 represent a noticeable improvement in the learners' performance over PLOT (see results in Bank, Crime, and German datasets).
4. The paper argues that using the debiased classifier for assigning pseudo-labels enables AdOpt to explore faster. Why do the authors not consider other RL methods(such as Double Deep-Q) as the baselines?
5. Why is MNIST selected as the comparative dataset while the paper examines the Bank Loan Problem?


**Summary Of The Paper:**

The paper studies a class of problems known as the Bank Loan Problem (BLP), where the learner only observes whether a customer will repay a loan if the loan is accepted. The labeled training data in this problem is biased since it is affected by previous decisions. The authors propose adversarial optimism (AdOpt) to address this problem utilizing adversarial domain adaptation. First, they train a debiased classifier using adversarial training and incorporate this classifier with the pseudo-label optimism (PLOT) method to increase the rate of correct decisions. The numerical results demonstrate the advantages of AdOpt in the standard BLP benchmark.


**Summary Of The Review:**

Overall, I find the paper interesting, and I appreciate this approach to solving an important but often neglected problem in ML.

---

> ### Author Response · Authors · 2022-11-10
> **Response to Reviewer ktkm**
>
> Thank you for taking time to carefully read our paper and recognizing the importance of the problem setup and novelty of our approach. We address the issues you raised below.
> 1. You wrote: *The use of adversarial domain adaptation in the paper is not conceivable. The adversarial domain adaptation aims to equalize the probability of 1s from both domains (Equation 5). However, it inherently contradicts the PLOT method (Step 5, Algorithm 1).*
> There is no contradiction here - the *combination* of adversarial domain adaptation with pseudo-label filtering **is** the core idea of our method. AdOpt applies the PLOT method to the optimistic (as in Equation 5) predictions of the adversarially trained classifier in step 5 of the Algorithm 1. The fundamental problem that AdOpt attempts to solve is that of the self-reinforcing false rejections. Let us recap how Algorithm 1 does it. The following explanation is an expanded version of the one in the begining of section 4.
> * Observe unlabelled batch B and obtain biased (i.e.trained only on accepted applicants data) and adversarially trained classifiers for previously accepted data and B.
> * If a data point is *rejected* by the biased classifier (our case of interest) we use the adversarial classifier to decide whether to add it to the pseudo-label dataset from PLOT.(this is step 4 of Algorithm 1). *It follows from Equation 5 and Table 2 that there will be many such points - adversarially trained classifier has high recall.*
> * We then apply the Pseudo-label mechanism from PLOT, i.e. retraining on optimistic lables, on these candidates (this is step 5 of Algorithm 1). *The addition of PLOT helps compensate for lowered precision of adversarially trained classifier in step 4.*
>
> * We accept those points that were recommended for acceptance in step 4 AND those that were recommended for acceptance by the biased classifier.
>
>     In the revision draft we have slightly augmented Algorithm 1 to make the idea that PLOT only operates on applicants rejected by biased classifier and accepted by AdOpt more transparent.
>
> 2. You wrote *It is unclear how the de-biased classifier's performance impacts the learners' final performance compared to the previous method (PLOT). It would be valuable if the authors could provide a visualisation of the proposed method (see Figure 2 in the PLOT paper).*
> The phenomenon illustrated in Figure 2 of Pacchiano et al is simply that the bigger training set of accepted applicants gets the more it influences the pseudo-label model decision compared to the pseudo-labelled batch. This phenomenon is true for the PLOT approach of learning with pseudo-labels *regardless* of the method of choosing pseudo-label candidates. Once the set of accepted applicants is big enough, adding pseudo-labelled batch no longer changes the decision boundary. Figure 2 in Pacchiano el al illustrates this point for the "toy" example of XOR dataset. *Edited* However, it is plausible that that such stabilisation of decision boundary should happen earlier for AdOpt than for PLOT. After considering your suggestion further we agree that a visualisation comparing the evolution of the boundary of PLOT and AdOpt might be valuable to illustrate this, perhaps for a dataset with more complicated decision boundary than XOR. We will produce such images and include them in the revised version.

---

> > ### Author Response · Authors · 2022-11-10
> > **Response to Reviewer ktkm**
> >
> > 3. You wrote *The proposed approach requires training an additional debiased classifier in comparison to PLOT. However, I am not convinced that the experimental results in Figure 2 represent a noticeable improvement in the learners' performance over PLOT (see results in Bank, Crime, and German datasets).*
> > Figure 2 reported the results of experimental evaluations run with the shortest possible duration of adversarial training of just 1 epoch for every batch. At this setting the runtime for AdOpt doesn't differ significantly from the runtime PLOT(e.g. running times were respectively 3379.1 sec and 4581.2 for PLOT and AdOpt on Adult, 3415.8 sec 4235.8 sec on MNIST, and similarly for other datasets). This mode of training confers AdOpt significant advantage on Adult and MNIST. To quantify the statistical significance we have computed the t-values at 2500 steps. The t-values averaged over 5 seeds with 5 runs for each were 7.6778 on Adult and 3.8547 on MNIST (Figure 2) signifying a very high degree of confidence. Generally for 25 experiments we considered t-values of >2 as a cut-off since it rejects the null hypothesis with 95 percent accuracy.
> > With 1 epoch of adversarial training per batch AdOpt ties with PLOT on the rest of the datasets: PLOT, AdOpt and NeuralUCB tie on Bank (PLOT vs AdOpt: -0.7147, NeuralUCB vs AdOpt 0.6602) and PLOT and AdOpt tie on Crime and German with t-values of  0.3842  and -0.6844. (Note that German and Crime are the two additional datasets that were not examined Pacchiano et al). Negative t-values are used where AdOpt exhibited higher average regret.
> > However **importantly** we note that with increasing the length of adversarial training at each batch to 10 epochs AdOpt shows statistically significant advantage over PLOT on Crime (t-value of 2.4451) and it's performance is now better than that of PLOT on German, with t-value indicating a trend towards approaching statistical significance (t -value of 1.6827 ). While this does somewhat increase runtime, compute has low priority for our purposes: compute is cheap while mistakes are costly. The results on German and Crime are in Figures 3 and 4 of the revision draft.
> >
> > 4. You wrote: *The paper argues that using the debiased classifier for assigning pseudo-labels enables AdOpt to explore faster. Why do the authors not consider other RL methods(such as Double Deep-Q) as the baselines?*
> > The setting we consider is a special case of a contextual bandit, so methods like Double DQN are not applicable since there is no target value. Instead, we do benchmark against methods appropriate for contextual bandits and experimentaly verify that we outperform them.
> >
> > 5. *Why is MNIST selected as the comparative dataset while the paper examines the Bank Loan Problem?*
> > MNIST is the classic benchmark dataset for Deep Learning and is easily convertible to the BLP setting as described in the paper. It was also used by the authors of PLOT, hence we wanted to compare the performance of our algorithm compared to theirs on this dataset.
> >
> > We are very happy to answer any additional comments you might have and hope to engage in a productive discussion!

---

> ### Author Response · Authors · 2022-11-14
> **To reviewer ktkm**
>
> Thanks again for reviewing our paper. We hope our comments clarified some of the issues you raised:. Could you please let us know if you have any additional questions or comments, so we could attempt to address them?

---

### Author Response · Authors · 2022-11-10
**Summary of changes in revision draft**

We have made the following changes in the revised draft:
* We have noticed and corrected a mistake in the computation of t-values in Section 5. t-values are now averaged over all runs and seeds in the experiments. We have also corrected associated mistakes in the reporting of the statistical significance of results on some datasets.
* We have added figures to further clarify the empirical regret cumulative regret results in response to questions from the reviewers **cz4a** and **ktkm**. Figure 2 now shows cumulative regrets on Adult, MNIST and Bank, and Figures 3 and 4 show cumulative regrets on Crime and German and their dependence on the length of adversarial training.
* We have added an explanation of the "biased"/"de-biased" terminology in Section 3.3 following suggestions of reviewer **4jYB**
* We have added ablation study of AdOpt vs pseudo-labelling using a biased classifier with lower thresholds as suggested by **4jYB** in Figure 6.
* We have amended Figure 5 illustrating comparative bias of different methods for better clarity.

---

### Decision · Program_Chairs · 2023-01-20

**Decision:**

Reject

**Justification For Why Not Higher Score:**

The proposed method is incremental with respect to prior work and it only shows small gains over previous baselines.

**Justification For Why Not Lower Score:**

N/A

**Metareview: Summary, Strengths And Weaknesses:**

This paper studies the bank-loan problem, a binary online problem where the learner only receive positive or negative reward when it classifies the sample as positive and 0 reward when it classifies it as negative. The aim of the learner is the maximize the reward obtained.

While the problem can be modeled as a contextual bandit problem, there is a line of recent papers proposing heuristic solutions based on deep learning. This submission falls in this category presenting a heuristic method that builds on top of Pacchiano et al. (2021). The scientific claim of the submission is that the proposed algorithms improves previous work. This claim is validated through an empirical evaluation over 5 datasets.

This submission has been thoroughly discussed by the reviewers and the authors, without reaching an agreement on the significance of the results. None of the reviewers fully accepted this paper, but the discussion focused on the statistical significance of the gains over the previous algorithms.
On one hand, it is (almost*) safe to claim that the t-values calculated by the authors show a statistical significant result on 2 datasets out of 5, while the results on the other 3 are less clear. However, there is a difference between "statistically significant results" and the "significance" that the reviewers are asked to evaluate in their forms. A result can be statistically significant, as in the results in this paper, yet not significative enough to warrant publication in a selective venue as ICLR. In this view, unfortunately I agree with Reviewer cz4a who points out that the gains over the other algorithms might be too small to be interesting for this community. Note that even other communities would have the same standard distinguishing the statistical significance from the size of the effect. Moreover, the incremental nature of the proposed solution and the lack of any theoretical justification make the contribution even smaller. For these reasons, the paper cannot be accepted in the current form at ICLR.

* the "almost" above refers to the fact that comparing multiple algorithms on multiple datasets is a tricky business. I would encourage the authors to read Demšar, "Statistical Comparisons of Classifiers over Multiple Data Sets", JMLR 2006 for an actual principled way to do it.